# DIVIDE, REWEIGHT, AND CONQUER: A LOGIT ARITHMETIC APPROACH FOR IN-CONTEXT LEARNING

## ABSTRACT

In-Context Learning (ICL) emerges as a key feature for Large Language Models (LLMs), allowing them to adapt to new tasks by leveraging task-specific examples without updating model parameters. However, ICL faces challenges with increasing numbers of examples due to performance degradation and quadratic computational costs. In this paper, we propose **L**ogit **A**rithmetic **R**eweighting **A**pproach (LARA), a novel framework that enhances ICL by using logit-based ensembling of multiple demonstrations. Our approach divides long input demonstrations into parallelizable shorter inputs to significantly reduce memory requirements, and then effectively aggregate the information by reweighting logits of each group via a non-gradient optimization approach. We further introduce Binary LARA (B-LARA), a variant that constrains weights to binary values to simplify the search space and reduces memory usage by filtering out less informative demonstration groups. Experiments on BBH and MMLU demonstrate that LARA and B-LARA outperform all baseline methods in both accuracy and memory efficiency. We also conduct extensive analysis to show that LARA generalizes well to scenarios of varying numbers of examples from limited to many-shot demonstrations. Our codes can be found in `https://anonymous.4open.science/r/LARA-F55B`.

## 1 INTRODUCTION

**I**n-**C**ontext **L**earning (ICL) (Brown et al., 2020) is one of the emergent abilities of Large Language Models (LLMs) as they are scaled to billions of parameters (Wei et al., 2022). ICL enables LLMs to adapt to new tasks by utilizing task-specific examples within the input context (Dong et al., 2023), and does not require any updates to or access to model parameters. While ICL has achieved impressive performance across various domains, it encounters significant challenges when dealing with an increasing number of examples. Longer context window size often leads to performance degradation (Xiong et al., 2023). This is due to the low density of useful information within longer prompts, and the reduced sensitivity to positional information, both of which diminish the capability of the model to effectively capture and utilize key content. Additionally, the quadratic growth of computational cost with the input length makes it particularly expensive for large-scale models.

Previous works primarily focus on two directions to address these challenges. The first direction is input compression, which aims to shorten the input length (Jiang et al., 2023b; Pan et al., 2024; Xu et al., 2023a; Wingate et al., 2022) or selectively retrieve relevant portions of demonstrations to be included in the prompt (an Luo et al., 2024). However, these methods risk losing critical information, which may negatively impact model performance. The second direction involves aggregating hidden states within LLMs to simulate the effect of in-context demonstrations (Hao et al., 2022; Li et al., 2023b; Hendel et al., 2023). These methods, however, are not applicable to closed-source models like GPT-4, as they require direct access to the model internal weights. Additionally, they contradict the core advantage of in-context learning, which is the ability to operate without modifications to hidden states or model parameters.

In this study, we propose a novel framework, **L**ogit **A**rithmetic **R**eweighting **A**pproach (LARA), which aims to combine the strengths of both input compression and hidden state approaches. Our method first divides demonstrations into subgroups to allow LLMs to focus on shorter inputs and reduce computational requirements. We then design a weighted sum aggregation approach to combine

Figure 1: Illustration of the differences between few-shot in-context learning and LARA (ours) during inference. Unlike few-shot in-context learning, which concatenates all demonstrations as a prefix to the input, our method splits the in-context examples into different groups. The next token is then generated based on a weighted average of logits, with weights precomputed using the framework described in Sec. 3.3.

the output logits from the language model given each subgroup of examples. This ensures that the relevant information from each subgroup could potentially be captured by the language model. One key innovation in LARA is that we use a non-gradient approach to optimize the weights of logits for each subgroup. We employ the Covariance Matrix Adaptive Evolution Strategy (CMA-ES) (Hansen & Ostermeier, 1996) to efficiently explore the weight vector space via resampling based on best-performing candidates. This allows us to optimize the contribution of each subgroup without any gradient updates. We further develop Binary-LARA (B-LARA) by constraining the weight values to $\{0, 1\}$, which can be interpreted as a process of subgroup selection. This not only reduces the computational cost but more importantly, leads to better performance due to the simplified search space for the binary weight vector.

Our experiments on BBH and MMLU benchmarks show that both LARA and B-LARA consistently outperform direct in-context learning and simple retrieval-based demonstration selection across various models, with the additional benefit of lower GPU memory usage. Further analysis reveals that the method excels in both low-resource scenarios with few examples and settings with abundant demonstrations, consistently delivering superior performance. Moreover, our ablation study highlights the critical role of the reweighting steps, although even logit averaging alone outperforms standard in-context learning.

To summarize, our main contributions are as follows:

- To the best of our knowledge, we are the first to propose ensembling information through logit arithmetic from different ICL demonstrations. We introduce LARA, a non-gradient optimization framework that reweights the information of different demonstration groups to improve ICL performance.

- We conduct extensive experiments on Llama3.1-8B (Dubey et al., 2024), Mistral-7B (Jiang et al., 2023a), and Gemma-7B (Mesnard et al., 2024) on BBH Srivastava et al. (2022) and MMLU Hendrycks et al. (2021), and show that LARA outperforms all baseline methods across all three models.

- Our comprehensive analysis reveals the broad applicability and efficiency of LARA and B-LARA. We demonstrate that our methods consistently outperform baselines across a wide range of example quantities, from fewer than 5 to more than 200. We also demonstrate the applicability of our methods to black-box LLMs.

## 2 PRELIMINARIES

**In-Context Learning.** Traditional In-Context Learning leverages $N$ labeled examples in the input prompt, represented as $\mathcal{D}_{\text{train}} = \{(\boldsymbol{x_i}, \boldsymbol{y_i})\}_{i=1}^{N}$ to provide hints for language model generation. Each pair $(\boldsymbol{x_i}, \boldsymbol{y_i})$ is converted into a semantically meaningful demonstration $d_i = \tau(\boldsymbol{x_i}, \boldsymbol{y_i})$ using a predefined template $\tau$. These demonstrations are then concatenated to form a comprehensive context $\mathcal{C} = d_1 \oplus d_2 \oplus \cdots \oplus d_N$, with appropriate separators (*e.g.*, newlines or special tokens) between

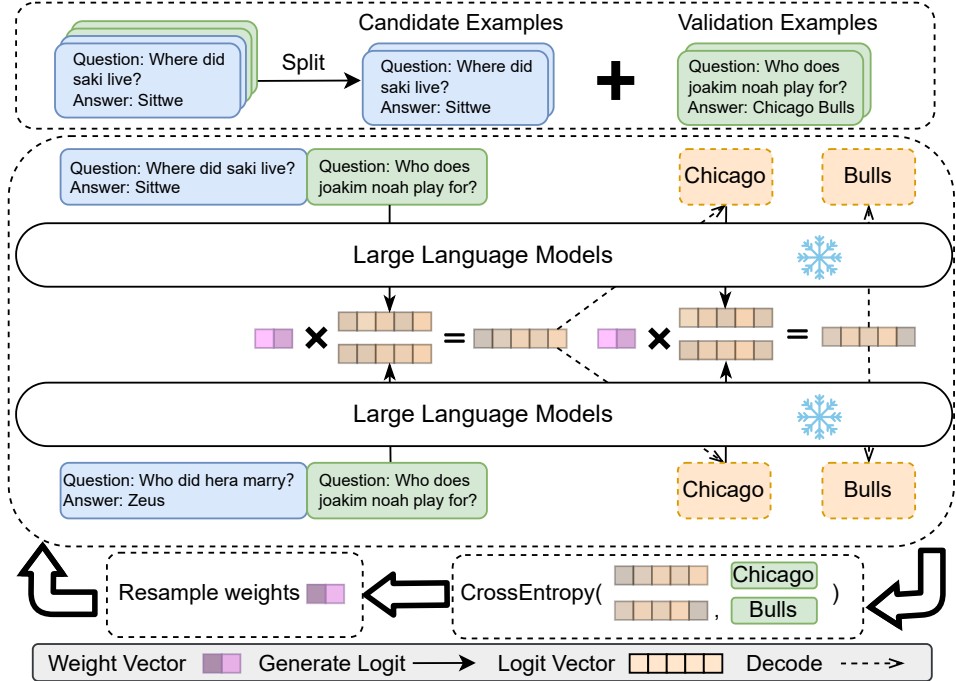

Figure 2: Illustration of the LARA framework. The input demonstration set $\mathcal{D}_{\text{train}}$ is divided into subsets $\mathcal{S}_1, \mathcal{S}_2, \ldots, \mathcal{S}_k$, which are further split into two groups: one for candidate examples and the other for validation examples. For each token, logits are generated using Logit-Arithmetic Decoding, which aggregates the output logits from all subsets. After generating all tokens, the cross-entropy loss is computed based on the weighted-average logits and the ground truth from the validation subset. The subset weights are then resampled and adjusted to minimize the loss. This process of token generation, loss calculation, and weight resampling is repeated iteratively. After optimizing the weights for the first group of candidate examples, the roles of the candidate and validation examples are swapped.

each demonstration. For each test input $\boldsymbol{x}_{\text{test}}$, the language model receives the concatenated prompt $\mathcal{C} \oplus \boldsymbol{x}_{\text{test}}$ to generate a response.

**Logit-based Generation.** We consider decoding approaches for language generation, where the language model receives an input prompt $\mathcal{C} \oplus \boldsymbol{x}_{\text{test}}$ and produces coherent and logical responses. The term "logit" refers to the raw, unnormalized scores output by the model before they are converted into probabilities by a softmax function. These logits are generated by passing the input sequence through the LLM. Formally, given the logit $\boldsymbol{z}$, the probability of the next token $x_t$ given the previous tokens $x_{1:t-1}$ is computed using the softmax function:

$$P(x_t \mid x_{1:t-1}) = \frac{\exp(\boldsymbol{z}_{x_t})}{\sum_{x' \in V} \exp(\boldsymbol{z}_{x'})} \tag{1}$$

where $\boldsymbol{z}_{x_t}$ is the logit corresponding to the token $x_t$, and $V$ is the vocabulary set.

## 3 METHODOLOGY

In this section, we provide an overview of LARA. Figure 2 illustrates the overall framework of our approach. Unlike directly concatenating $\mathcal{D}_{\text{train}}$ into a single sequence, we first divide the $N$ examples into subgroups, which are used as inputs to the LLM. The output logits from these subgroups are then aggregated, and we assign weights to each subgroup using a non-gradient search algorithm. During inference, the precomputed weights are used to combine the logits from each group.

In Sec. 3.1, we explain the partition strategy to divide examples into subgroups. Then we introduce how the outputs are aggregated across different subgroups in Sec. 3.2, and the reweighting strategy for optimal combination in Sec. 3.3. Furthermore, we show in Sec. 3.4 that imposing a hard constraint for our reweighting strategy could further reduce memory usage and computational resources. Finally, we discuss in Sec. 3.5 the inference efficiency brought by our proposed approach.

## 3.1 PARTITION STRATEGY

Given $N$-shot in-context examples, we first split $\mathcal{D}_{\text{train}}$ into $k$ disjoint subsets each containing $L$ in-context examples, such that $\mathcal{D}_{\text{train}} = \mathcal{S}_1 \cup \mathcal{S}_2 \cup \ldots \cup \mathcal{S}_k$ with $|\mathcal{S}_i| = L$ for all $i \in \{1, \ldots, k\}$. When inputting a subgroup $\mathcal{S}_i$ to an LLM, we concatenate all of its elements to get $\mathcal{C}_i = d_{(i-1)L+1} \oplus d_{(i-1)L+2} \oplus \cdots \oplus d_{iL}$, and the complete input for the $i$-th subgroup to LLM is $\mathcal{C}_i \oplus \boldsymbol{x}_{\text{test}}$. We assume that $N$ is divisible by $k$ in our experiments, so that $L = N/k$. In practice, in cases where $N$ is not divisible by $k$, we could truncate the last subset and only retain $L(k-1)$ examples.

## 3.2 LOGIT-ARITHMETIC DECODING

Previous studies (Li et al., 2022; Liu et al., 2024; Dekoninck et al., 2023) have utilized logit offsets to control the outputs of large language models for better generation quality or instruction following. Inspired by these work, we propose a novel method that combines information from multiple in-context demonstrations through logit-arithmetic decoding. Specifically, our approach focuses on aggregating the logits produced by the language model outputs for various contextual inputs. With the input query $\boldsymbol{x}_{\text{test}}$ and the example subset being $\mathcal{S}_i$, we can compute the logit outputs of the language model, denoted as $f_\theta(\mathcal{S}_i, \boldsymbol{x}_{\text{test}}) = \log p(y \mid \mathcal{S}_i, \boldsymbol{x}_{\text{test}})$. We then combine these logits using a weighted sum to get the generation probability over the output token:

$$p(y \mid \boldsymbol{x}_{\text{test}}, \boldsymbol{w}) = \text{softmax}\left(\sum_{i=1}^{k} w_i \cdot f_\theta(\mathcal{S}_i, \boldsymbol{x}_{\text{test}})\right) \tag{2}$$

where $k$ is the number of example subsets, and $w_i$ are weights that indicate the importance of the contribution of each subset, with $\sum_{i=1}^{k} w_i = 1$. As a baseline approach, we could set uniform weighting, where $w_i = 1/k$. However, this may not be optimal for all tasks, as the quality and relevance of different subgroups may vary. In the following section, we introduce a reweighting strategy to optimize these weights to enhance model performance.

## 3.3 REWEIGHTING LOGITS BY NON-GRADIENT OPTIMIZATION

To further enhance the model performance, we employ non-gradient optimization methods to optimize the weights $w_i$ based on the loss calculated from $p(y \mid \boldsymbol{x}_{\text{val}})$. Given the combined probability $p(y \mid \boldsymbol{x}_{\text{val}})$, our objective is to minimize a cross-entropy loss function $\mathcal{L}(\boldsymbol{w})$ over the predicted probabilities and the ground truth. Specifically, we utilize the following cross-entropy loss function for the generation model:

$$\mathcal{L}(\boldsymbol{w}) = - \sum_{(\boldsymbol{x}_{\text{val}}, \boldsymbol{y}_{\text{val}}) \in \mathcal{D}} \sum_{t=1}^{T} \log p(y_t \mid \boldsymbol{x}_{\text{val}}, \boldsymbol{w})$$

where $D$ represents the validation dataset, $T$ is the length of the sequence, $y_t$ is the true word at time step $t$, $\boldsymbol{x}_{\text{val}}$ is the input sequence, $\boldsymbol{w}$ denotes the weight vector, and $p(y_t \mid \boldsymbol{x}_{\text{val}}, \boldsymbol{w})$ represents the predicted probability of the true word $y_t$ at time step $t$, given the input sequence $\boldsymbol{x}_{\text{val}}$ and the weight vector $\boldsymbol{w}$.

To avoid introducing additional labeled data, we employ a cross-validation strategy. We partition the demonstration set $\mathcal{S}$ into two subsets: $\mathcal{S}_A = \mathcal{S}_1 \cup \mathcal{S}_2 \cup \ldots \cup S_{\lfloor k/2 \rfloor}$ and $\mathcal{S}_B = \mathcal{S}_{\lfloor k/2 \rfloor + 1} \cup \mathcal{S}_{\lfloor k/2 \rfloor + 2} \cup \ldots \cup \mathcal{S}_k$. When optimizing weights for $\mathcal{S}_i \in \mathcal{S}_A$, we use $\mathcal{S}_B$ as the validation set, and vice versa.

We choose non-gradient optimization methods over gradient-based alternatives due to two key factors: (1) The loss function $\mathcal{L}(\boldsymbol{w})$ is non-differentiable, since updating the weight vector $\boldsymbol{w}$ affects the logits of subsequent tokens, leading to possibly different decoding results of subsequent tokens. (2) The dimensionality of the weight vector $\boldsymbol{w}$ is relatively low, specifically equalled to the number of groups $k$.

In our empirical experiments, we refer to Liu et al. (2020) and employ the Covariance Matrix Adaptive Evolution Strategy (CMA-ES) (Hansen & Ostermeier, 1996). CMA-ES is a stochastic, derivative-free optimization algorithm. During each iteration, CMA-ES samples a set of candidates in the space of the weight vector $\boldsymbol{w}$ from a multivariate normal distribution, evaluates $\mathcal{L}(\boldsymbol{w})$ for each candidate, and then updates the mean and covariance matrix of the distribution based on the best-performing candidates. This allows for an efficient exploration over the weight space.

### 3.4 BINARY CONSTRAINTS FOR LARA

We further propose a variant of LARA, named as B-LARA, by imposing a hard constraint on the weight vector $\boldsymbol{w}$ to binary values $\{0, 1\}$. This binary constraint offers two key advantages: first, it simplifies the search space and potentially leads to faster convergence; second, it allows for direct elimination of demonstration groups with zero weight, thereby improving inference efficiency. Intuitively, the binary optimization of $\boldsymbol{w}$ can be seen as a form of subset selection to identify the most relevant demonstrations in $\mathcal{D}_{\text{train}}$ benefitting model performance on specific tasks.

To solve this binary optimization problem, we employ the simplest evolution strategy (1+1)-ES (Rechenberg, 1973). It involves a simple cycle: a single parent produces one offspring per generation through mutation—adding a small, random change. If this offspring performs as well or better than the parent based on a predefined fitness criterion, it becomes the new parent for the next generation. Otherwise, the original parent remains. The overall sampling procedure is shown in Algorithm 1.

The simplicity of this method in repeated mutation and selection makes it particularly suitable for our binary optimization scenario.

### 3.5 COMPUTATIONAL COMPLEXITY

We analyze the computational complexity of LARA and B-LARA compared to standard ICL. During inference, the self-attention mechanism in Transformer models is the primary bottleneck for GPU memory requirement, with the memory complexity being $O(n^2)$, where $n$ is the input sequence length. This quadratic scaling is due to the pairwise interactions between tokens in the attention matrix.

By splitting the input sequence into $k$ groups, each of length around $\frac{n}{k}$, LARA and B-LARA can leverage parallel computing resources more effectively. The complexity for LARA becomes $O(\frac{n}{k}^2 * k) = O(\frac{n^2}{k})$. B-LARA further reduces computational complexity by selecting only a subset of groups. If $m$ out of $k$ subgroups are assigned non-zero weights, then the complexity of B-LARA becomes $O(\frac{mn^2}{k^2})$. We show the empirical GPU memory usage in Sec. 5.5.

## 4 EXPERIMENTS

In this section, we provide details of our main experiments. We first give an overview of the experimental setup and implementation details in Sec. 4.1, and then present our findings along with the results in Sec. 4.2.

### 4.1 EXPERIMENTAL SETUP

**Datasets and Evaluation.** We evaluate our methods using two well-established benchmarks: Big-Bench Hard (BBH) (Srivastava et al., 2022) and Massive Multitask Language Understanding

(MMLU) (Hendrycks et al., 2021). BBH tests models on challenging reasoning tasks across domains including arithmetic reasoning, commonsense reasoning, and linguistics. MMLU measures generalization across 57 diverse subjects, covering both humanities and STEM fields, offering a comprehensive evaluation of knowledge and problem-solving abilities of LLMs. For both benchmarks, we use exact match (EM) as our evaluation criterion, which requires model predictions to perfectly match the correct answers. We report the accuracy scores in our experiment results. The details about dataset analysis and prompts can be found in Appendix A.

**Models.** Our proposed LARA for in-context learning is applicable to any LLM. To demonstrate its generality, we evaluate it on three open-source, decoder-only models: Llama3.1-8B (Dubey et al., 2024), Mistral-7B (Jiang et al., 2023a), and Gemma-7B (Mesnard et al., 2024). Llama-3.1-8B is known for strong performance across various NLP tasks, Mistral-7B is optimized for efficiency and is balanced between computational cost and accuracy. Gemma-7B focuses on advanced reasoning and language comprehension. These models represent diverse architectures and training strategies, allowing us to test the adaptability of our methods. By using open-source models in evaluation, we ensure the reproducibility of our proposed method and validate its broad applicability across state-of-the-art model architectures.

**Hyperparameter Setting.** In our main experiment, we use $\mathcal{D}_{\text{train}}$ consisting of $N = 32$ in-context examples for our methods. For each task, $\mathcal{D}_{\text{train}}$ is split into subsets of size $L \in \{2, 4, 8\}$, and for each $L$ we perform up to $J = 20$ iterations for weight optimization. We compare the minimum validation loss across different settings of $L$ to determine the optimal configuration, including $L$ and corresponding $w$, for the final inference phase. The baseline methods also use the same $\mathcal{D}_{\text{train}}$ as input. For our method and all baselines, we set the temperature to 0 to enforce greedy decoding. Our experiments are conducted on a single A100 80GB GPU.

**Compared Methods.** We introduce several primary baseline methods: Direct In-Context Learning (ICL), KNN-Augmented In-ConText Example Selection Liu et al. (2022) (KATE), Rationale-Augmented Ensembles (RAE) (Wang et al., 2022) and In-context Vector (ICV) (Liu et al., 2023) and StructICL (Hao et al., 2022) as the representative of parameter access methods. We use the same 32 in-context examples as inputs to all baseline methods as our proposed method. For Direct ICL, all 32 examples are concatenated with the prompt. For KATE, we apply the Top-K selection from Liu et al. (2022) that uses a smaller model[1] to retrieve the most similar input-output pairs from $\mathcal{D}_{\text{train}}$ as in-context demonstrations. We evaluate KATE with 2, 4, and 8 demonstrations as baselines. For RAE, we divide the examples into different groups and use each group as in-context examples to generate separate results. The final output is determined by applying majority voting across these individual group-based results. For StructICL, we also present the results with varying numbers of groups: 2, 4, and 8. In ICV, we follow the original paper to set $\lambda = 0.1$ and average the ICV given by all 32 examples. We report results with group sizes of 2, 4, and 8 to ensure the same memory usage as our method.

## 4.2 MAIN RESULTS

Results from Table 1 demonstrate the effectiveness of our proposed methods, LARA, and B-LARA, across BBH and MMLU benchmarks. B-LARA consistently outperforms most of baseline methods across three model architectures. Notably, B-LARA achieves the highest accuracy and improves over direct ICL by 2.05, 5.67, and 2.12 points on BBH dataset across three models respectively. Moreover, our methods and can consistently outperform retrieval or simple ensemble baselines like KATE and RAE, indicating that our method is more effective in combining information from multiple demonstration subgroups. Compared to the ICV and StructICL baseline, which has the advantage of access to model parameters, our methods still achieve better performance without access to the hidden state, which further demonstrates the efficacy of our methods in aggregating information without direct access to model internal parameters.

An interesting finding is that B-LARA performs better than LARA despite a more constrained search space for the weight vector. We believe this is because we only use 20 iterations for weight opti-

---

[1]https://huggingface.co/sentence-transformers/all-distilroberta-v1

Table 1: Accuracy of all methods on BBH and MMLU. The results shown are the average performance across datasets within each benchmark. Please refer to appendix D.2 for breakdown results of each dataset. The subscript of KATE indicates the number of selected ICL demonstrations as input to LLMs.

| | $BBH_{average}$ | | | $MMLU_{average}$ | | |
|---|---|---|---|---|---|---|
| | Llama3.1-8B | Gemma-7B | Mistral-7B | Llama3.1-8B | Gemma-7B | Mistral-7B |
| *Black-Box Method:* | | | | | | |
| ICL | 45.64 | 37.08 | 42.91 | 65.63 | 61.44 | 62.84 |
| $KATE_2$ | 43.60 | 37.07 | 43.16 | 66.62 | 56.28 | 63.99 |
| $KATE_4$ | 44.03 | 38.83 | 43.16 | 66.75 | 55.78 | 63.48 |
| $KATE_8$ | 44.47 | 37.03 | 42.96 | 67.19 | 54.13 | 63.93 |
| $RAE_2$ | 44.59 | 40.24 | 43.95 | 66.88 | 65.18 | 62.99 |
| $RAE_4$ | 45.23 | 40.44 | 44.49 | 66.40 | 65.01 | 62.99 |
| $RAE_8$ | 44.06 | 39.85 | 44.07 | 67.09 | 64.80 | 63.61 |
| LARA (ours) | 47.46 | 41.77 | 44.77 | 66.54 | 64.36 | 63.93 |
| B-LARA (ours) | **47.69** | **42.75** | **45.03** | **67.80** | **65.56** | **64.12** |
| *White-Box Method:* | | | | | | |
| ICV | 45.93 | 42.16 | 44.50 | 66.97 | 64.99 | 64.02 |
| $StructICL_2$ | 46.64 | 39.54 | 44.68 | 66.78 | 64.34 | 63.52 |
| $StructICL_4$ | 46.98 | 40.53 | 44.89 | 66.97 | 64.46 | 63.99 |
| $StructICL_8$ | 46.57 | 41.46 | 43.99 | 66.56 | 65.16 | 63.46 |

mization, and the binary constraint brings more benefits by introducing a simplified optimization landscape and providing a regularization effect to prevent overfitting.

# 5 ANALYSIS

In this section, we present a deep analysis of our proposed method LARA under various conditions.

## 5.1 HOW DOES LARA EXTEND TO GENERATION TASKS?

In previous experiments, we mainly focus on the classification or single token generation tasks. Here we extended our experiments to generation tasks like GSM8K (Cobbe et al., 2021) for math reasoning. We follow the experiment setting used in FocusICL (Yuan et al., 2024) and evaluate our method against Llama-3-8B-Instruct, LongChat-7B-V1.5-32K (Li et al., 2023a) and Vicuna-7B-V1.5-16K (Dong et al., 2023). Following FocusICL, we randomly select 80 examples from the training set and split them into 10 groups for our methods.

Table 2: Accuracy across different models on Gsm8k.

| | Llama-3 | LongChat | Vicuna |
|---|---|---|---|
| ICL | 66.64 | 9.93 | 16.30 |
| EarlyStop | 71.21 | 11.14 | 17.44 |
| StructICL | 69.43 | 11.25 | 17.12 |
| FocusICL | 71.89 | **12.28** | 17.74 |
| B-LARA | **73.86** | 12.23 | **18.12** |

The results in Table 2 show B-LARA outperforms FocusICL, which is the previous state-of-the-art method, in 2 out of 3 models. Notably, this is achieved without relying on hidden states, highlighting the simplicity and efficiency of our method on generation task.

## 5.2 CAN LARA PERFORM WELL WITH MORE EXAMPLES?

We investigate the performance of LARA with an increased number of demonstrations, leveraging the LongICLBench (Li et al., 2024), a benchmark tailored for addressing challenges in long in-context learning. For our experiments, we select two datasets: GoEmotion and TacRED. Following the LongICLBench setup, we employ multiple rounds of examples, where each round includes several examples, each labeled with a distinct class. To align with the input limit constraints of ICL, we sampled 8 rounds (224 examples) of examples for GoEmotions and 4 rounds (164 examples) for TacRED. For LARA and B-LARA, we choose 4, 8, and 16 as the potential candidate number of groups. We report the accuracy of different methods on these datasets in Table 3.

Table 3: Accuracy of methods on GoEmotion and TacRED. The subscript of RAE means the number of groups in RAE.

| | GoEmotion | | | TacRED | | |
|---|---|---|---|---|---|---|
| | Llama3.1-8B | Gemma-7B | Mistral-7B | Llama3.1-8B | Gemma-7B | Mistral-7B |
| *Black-Box Method:* | | | | | | |
| ICL | 18.60 | 15.60 | 17.80 | 38.20 | 43.80 | 55.40 |
| $RAE_2$ | 22.20 | 22.20 | 21.60 | 43.80 | 45.40 | 55.40 |
| $RAE_4$ | 21.00 | 22.40 | 21.40 | 45.60 | 45.00 | 52.40 |
| $RAE_8$ | 21.20 | 19.00 | 20.40 | 36.20 | 39.00 | 49.20 |
| LARA (ours) | 21.00 | 20.80 | 19.20 | **48.60** | 47.40 | 54.20 |
| B-LARA (ours) | **24.00** | **22.80** | **23.80** | **48.60** | **49.00** | **59.00** |
| *White-Box Method:* | | | | | | |
| ICV | 18.80 | 20.80 | 18.40 | 44.40 | 46.80 | 54.40 |
| $StructICL_2$ | 19.00 | 21.00 | 18.60 | 46.60 | 47.60 | 55.80 |
| $StructICL_4$ | 19.80 | 21.40 | 19.00 | 47.60 | 48.00 | 56.40 |
| $StructICL_8$ | 20.60 | 22.00 | 19.80 | 44.80 | 48.60 | 56.20 |

The experimental results clearly highlight the advantages of LARA, which demonstrates consistent improvements over baseline methods across both GoEmotion and TacRED datasets, showcasing its effectiveness in diverse tasks. Notably, the B-LARA variant further amplifies this performance, outperforming all competing approaches on both datasets and across various models. This suggests that B-LARA can work well in many shot settings.

## 5.3 CAN LARA PERFORM WELL WITH LIMITED IN-CONTEXT EXAMPLES?

In previous experiments, we primarily explore the many-shot in-context learning (ICL) setting. In this subsection, we focus on a more constrained scenario, where only a limited number of in-context examples are available. This analysis aims to understand the relationship between the number of demonstrations and the performance of LARA compared to baseline methods with limited examples.

We set the number of examples $N$ within $\{2, 4, 8, 16\}$ and compare our proposed method with ICL on the BBH dataset with Mistral-7B. Figure 3 demonstrates that both LARA and B-LARA consistently outperform the baseline ICL, and the performance gap increases with the number of examples used. Note that we do not plot the performance of LARA and B-LARA under $N = 2$. This is because LARA and B-LARA are simplified to our non-reweighting ablation when the size of each subgroup becomes 1 and no reweighting is required. We also show the performance of performance without reweighing here. We set the number of group $k$ as 2 in this experiment. While there is a significant gap between the non-reweight version and B-LARA, the non-reweight version still demonstrates effectiveness compared to ICL.

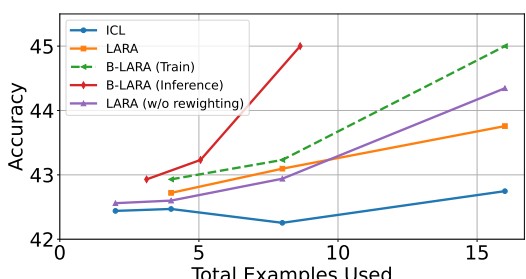

Figure 3: Accuracy of LARA on BBH using different numbers of examples. B-LARA uses different settings due to differences in example usage during training and inference. We use two lines to highlight this difference. The accuracy means the average accuracy on BBH dataset.

Since B-LARA has a weight constraint of $\{0, 1\}$, subgroups with zero-weights are pruned during inference for efficiency. As shown in Figure 3, the real number of examples used by B-LARA in inference is substantially lower than other methods. In the 32-shot setting, only about 45% of subgroups of B-LARA are assigned non-zero weights, reducing more than half of the computational load without compromising performance. Additionally, as the total number of examples increases, the proportion of examples used in inference decreases, indicating that B-LARA is particularly suitable for resource-constrained environments.

### 5.4 Is LARA Applicable to Black-Box LLMs?

One advantage of our method is that it could also be applied to LLM APIs, since it only uses output logits for example reweighting or selection. In these scenarios, techniques such as in-context vector or task vector, which often rely on internal state visibility, cannot be applied.

Table 4: Average performance of various methods of GPT-4o-mini on the BBH benchmark.

| ICL | LARA | B-LARA |
|---|---|---|
| 53.17 | 56.06 | 57.41 |

We evaluate our method with GPT-4o-mini [2] on BBH dataset. The results in Table 4 demonstrate that LARA and B-LARA outperform ICL. We note that the OpenAI API only provides top 20 logits for each output token, while our methods are still able to achieve competitive results. This indicates that our method generalizes well to black-box LLMs, and can be applied to situations where internal weights of models are restricted and only output logits are available.

### 5.5 How Does LARA Enhance Memory Efficiency?

We empirically evaluate the computational efficiency of LARA by measuring GPU memory usage with different input sequence lengths and subgroup configurations. We set the number of groups $k$ with 1,2,4,8. Specifically, when $k$ is set as 1, LARA will degrade to ICL.

Results in Figure 4 demonstrate that LARA is more memory-efficient compared to standard ICL, especially when handling long sequences. Standard ICL results in Out-of-Memory (OOM) errors when the input length exceeds 10k tokens on a Mistral-7B model with a batch size of 4 on an A100 80GB GPU. In contrast, our method handles input lengths over 25k tokens with 4 and 8 subgroups, demonstrating that LARA efficiently utilizes larger amounts of training data.

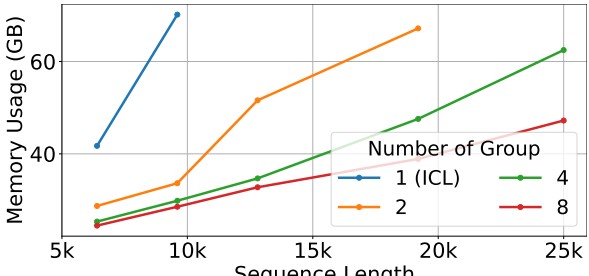

Figure 4: GPU Memory usage of LARA in gigabytes on a single A100 80GB GPU with different input sequence lengths and number of subgroups. Note that when the number of subgroups equals to 1, the setting is the same as ICL. The sequence length is denoted in thousands of tokens. We set the batch size equal to 4. Data points indicating Out-Of-Memory (OOM) are omitted.

### 5.6 How does the Reweighting Step Affect Model Performance?

We conduct an ablation study to assess the effectiveness of the reweighting step, denoted as "w/o reweight" which simply averages over the output logits of the LLM across different demonstration groups.

In our ablation study, removing the reweighting step used in LARA also demonstrated its value by outperforming traditional baseline methods. For instance, it achieved a notable 67.58 with Llama3.1-8B in the MMLU benchmark, which is better than directly ICL (65.63). This perfor-

Table 5: Average performance of Llama3.1-8B our methods without reweighting. For the ablation "w/o reweight", the subscript means the size $L$ of each group of demonstrations. The results for other models are shown in Appendix D.1

| Method | BBH | MMLU |
|---|---|---|
| LARA | 47.46 | 66.54 |
| B-LARA | **47.69** | **67.80** |
| w/o reweight$_2$ | 43.33 | 67.23 |
| w/o reweight$_4$ | 44.50 | 67.58 |
| w/o reweight$_8$ | 43.02 | 67.43 |

mance highlights that logit-arithmetic can successfully combine the information in different groups of demonstrations.

The results further emphasize the importance of the reweighting step in LARA. LARA outperforms the non-reweight version in most settings. This underscores the reweighting process as critical for

---

[2]gpt-4o-mini-2024-07-18

enhancing model accuracy. The worse performance of non-reweight offers clear evidence of how significant reweighting is to optimizing the model's contextual handling.

# 6 RELATED WORK

## 6.1 LONG IN-CONTEXT LEARNING

Recent studies on long-context learning problems in LLMs can be categorized into two main strategies: enhancing the impact of in-context examples and compressing input sequences. Structured prompting leverages rescaled attention mechanisms to effectively integrate grouped examples (Hao et al., 2022). Methods such as task vectors (Hendel et al., 2023) and function vectors (Todd et al., 2023) further refine this strategy by generating vectors that assess the contribution of each example based on the offset of hidden state, which improves model adaptability. Liu et al. (2023) generate task-specific vectors that steer model behavior in latent space based on the in-context examples. Regarding input compression, methods like prompt pruning (Jiang et al., 2023b; Pan et al., 2024) and additional summarization models (Xu et al., 2023a; Gilbert et al., 2023) directly shorten inputs while maintaining essential content. Soft prompt-based compression (Wingate et al., 2022; Mu et al., 2023) intends to generate a soft-prompt that includes most of the information. For many-shot in-context learning problems, previous studies have proposed group-based methods, such as StructICL (Hao et al., 2022) and FocusICL (Yuan et al., 2024), which refine attention maps by utilizing subgroup structures within the demonstrations.

## 6.2 LOGIT ARITHMETIC

Several works have employed logit arithmetic across various domains and downstream tasks. Contrastive decoding (Li et al., 2022) improves performance by utilizing the difference in logits from models of different sizes. Proxy tuning (Liu et al., 2024) enhances a larger model's capabilities by adding the logit differences of a smaller model, recorded before and after training, to simulate training effects. In model arithmetic (Dekoninck et al., 2023), logits adjusted with various prompts steer the generation processes of large language models. Huang et al. (2024) propose using logit subtraction to facilitate the selective forgetting of knowledge in LLMs. Additionally, logit arithmetic has been leveraged to enhance the safety of generated outputs (Xu et al., 2024).

## 6.3 NON-GRADIENT OPTIMIZATION OF LLMS

Due to the high memory requirements associated with gradient-based optimization methods, recent research has shifted towards non-gradient techniques for neural network optimization. Zhang et al. (2024); Malladi et al. (2023) propose training large language models (LLMs) using non-gradient methods to mitigate these memory constraints. These approaches have also been applied in federated learning, exploring their effectiveness in distributed settings (Xu et al., 2023b). Additionally, a gradient-free method has been used to optimize manifold neural networks (Zhang et al., 2022). Similarly, LoraHub (Huang et al., 2023) utilizes non-gradient techniques to dynamically reweight different LoRA modules, enhancing adaptation to new downstream tasks. Guo et al. (2023) also introduces non-gradient methods to prompt engineering to search for better prompts.

# 7 CONCLUSION

We proposed LARA, a novel framework that enhances in-context learning by ensembling logits from multiple demonstrations, improving performance without requiring parameter updates. Our method reduces computational complexity while achieving better accuracy. Additionally, Binary LARA further optimizes efficiency by selectively removing less informative demonstrations. Experiments on BBH and MMLU benchmarks show that both LARA and B-LARA outperform traditional ICL methods in terms of efficiency and performance. Future research directions include extending our study to combine logits from different sources beyond just in-context learning (ICL) examples—such as different models or varying instructions—and building a distributed inference system based on LARA.

REPRODUCIBILITY STATEMENT

We provide detailed information on the dataset we used in Appendix A. The codes for our main experiment can be found in `https://anonymous.4open.science/r/LARA-F55B`.

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

# A  DATASET DETAILS

## A.1  PROMPTS FOR INFERENCE

Table 6: Prompt examples for each dataset in One-shot learning.

| Dataset | Prompt |
|---------|--------|
| BBH | Question: {*question*}
Answer: {*answer*}

Question: {*question*}
Answer: |
| MMLU | The following are multiple choice questions (with answers) about {*subject*}.

Question: {*question*} Answer: {*answer*}

Question: {*question*} Answer: |
| GoEmotion | Given a comment, please predict the emotion category of this comment. The predict answer must come from the demonstration examples with the exact format.
The examples are as follows:
comment: {*question*}
emotion category: {*answer*}
comment: {*question*}
emotion category: |
| TacRED | Given a sentence and a pair of subject and object entities within the sentence, please predict the relation between the given entities.
You can only select from the following words: {*potential relation*}
sentence: {*question*}
the relation between the two entities is: {*answer*}
sentence: {*question*}
the relation between the two entities is: |

## A.2  DATASET STATISTICS

Table 7: Dataset Statistics.

| Dataset | #Tokens/Shot | Description |
|---------|--------------|-------------|
| BBH | 55 | A collection of challenging tasks from the BIG-Bench Hard benchmark. |
| MMLU | 65 | Multiple-choice questions across various subjects. |
| GoEmotion | 28 | Annotated Reddit comments for emotion classification. |
| TacRED | 80 | A dataset for relation extraction tasks. |

Table 8: Performance Comparison Across Models and Methods

| Model | Llama-3-8B-Instruct | | LongChat-7B-V1.5-32K | | Vicuna-7B-V1.5-16K | |
|---|---|---|---|---|---|---|
| | ARC | GSM8K | ARC | GSM8K | ARC | GSM8K |
| **ICL** | 90.00 | 66.64 | 62.43 | 9.93 | 77.11 | 16.30 |
| **EarlyStop** | 90.47 | 71.21 | 62.43 | 11.14 | 78.14 | 17.44 |
| **StructICL** | 90.70 | 69.43 | 64.05 | 11.25 | 78.05 | 17.12 |
| **FocusICL** | **91.02** | *71.89* | **64.55** | **12.28** | *78.51* | *17.74* |
| **B-LARA** | *90.89* | **73.86** | *64.27* | *12.23* | **78.79** | **18.12** |

---

**Algorithm 1** B-LARA Optimization Algorithm with Updated Index

---

**Input:** $\mathcal{D}_{\text{train}}$: In-context examples $\mathcal{D}_{\text{train}} = \{(\boldsymbol{x_i}, \boldsymbol{y_i})\}_{i=1}^{N}$.
**Parameter:** $k$: Number of subgroups. $J$: Number of iterations.
**Output:** $\boldsymbol{w}^*$: Optimized binary weight vector.
Split $\mathcal{D}_{\text{train}}$ into $k$ groups: $\{\mathcal{S}_1, \mathcal{S}_2, \ldots, \mathcal{S}_k\}$ $\mathcal{S}_A \leftarrow \{\mathcal{S}_1, \ldots, \mathcal{S}_{\lfloor k/2 \rfloor}\}$ $\mathcal{S}_B \leftarrow \{\mathcal{S}_{\lfloor k/2 \rfloor+1}, \ldots, \mathcal{S}_k\}$
**for** $r \in \{A, B\}$ **do**
    Initialize $\boldsymbol{w}^{(0)}$ as a random binary vector of length $|\mathcal{S}_r|$
    **for** $j = 1$ *to* $J$ **do**
        **for** $m = 1$ *to* $dim(\boldsymbol{w}^{(j-1)})$ **do**
            $u_m \leftarrow \text{Uniform}(0, 1)$
            $w'_m \leftarrow w_m^{(j-1)} \oplus \mathbb{I}(u_m < 1/\text{dim}(\boldsymbol{w}^{(j-1)}))$
        **end**
        Compute $\mathcal{L}(\boldsymbol{w}')$ using $\mathcal{S}_{r'}$, where $r' \neq r$
        **if** $\mathcal{L}(\boldsymbol{w}') \leq \mathcal{L}(\boldsymbol{w}^{(j-1)})$ **then**
            $\boldsymbol{w}^{(j)} \leftarrow \boldsymbol{w}'$
        **else**
            $\boldsymbol{w}^{(j)} \leftarrow \boldsymbol{w}^{(j-1)}$
        **end**
    **end**
    $\boldsymbol{w}_r^* \leftarrow \boldsymbol{w}^{(J)}$
**end**
$\boldsymbol{w}^* \leftarrow [\boldsymbol{w}_A^*, \boldsymbol{w}_B^*]$
**return** $\boldsymbol{w}^*$

---

# B COMPARISION TO PREVIOUS METHODS

# C ALGORITHM

# D FULL RESULTS

## D.1 FULL ABLATION STUDY

## D.2 FULL MAIN RESULTS

Here we will show the full results of our three models in BBH and MMLU benchmark. The methods include LARA, B-LARA, KATE, ICL, LAG(logit-average-generation which is the ablation study in our paper, together with RAE and ICV.

Table 9: Ablation Study Results

| | BBH$_{average}$ | | | MMLU$_{average}$ | | |
|---|---|---|---|---|---|---|
| | Llama3.1-8B | Gemma-7B | Mistral-7B | Llama3.1-8B | Gemma-7B | Mistral-7B |
| ICL | 45.64 | 37.08 | 42.91 | 65.63 | 61.44 | 62.84 |
| KATE$_2$ | 43.60 | 37.07 | 43.16 | 66.62 | 56.28 | 63.99 |
| KATE$_4$ | 44.03 | 38.83 | 43.16 | 66.75 | 55.78 | 63.48 |
| KATE$_8$ | 44.47 | 37.03 | 42.96 | 67.19 | 54.13 | 63.93 |
| RAE$_2$ | 44.59 | 40.24 | 43.95 | 66.88 | 65.18 | 62.99 |
| RAE$_4$ | 45.23 | 40.44 | 44.49 | 66.40 | 65.01 | 62.99 |
| RAE$_8$ | 44.06 | 39.85 | 44.07 | 67.09 | 64.80 | 63.61 |
| ICV | 45.93 | 42.16 | 44.50 | 66.97 | 64.99 | 64.02 |
| LARA | 47.46 | 41.77 | 44.77 | 66.54 | 64.36 | 63.93 |
| B-LARA | **47.69** | 42.75 | **45.03** | **67.80** | 65.56 | **64.12** |
| w/o reweight$_2$ | 43.33 | **43.56** | 42.83 | 67.23 | 65.61 | 62.95 |
| w/o reweight$_4$ | 44.50 | 41.98 | 44.78 | 67.58 | **65.87** | 63.32 |
| w/o reweight$_8$ | 43.02 | 39.35 | 44.84 | 67.43 | 65.04 | 63.55 |

| Task Name | LARA | B-LARA | KATE$_2$ | KATE$_4$ | KATE$_8$ | ICL | LAG$_2$ | LAG$_4$ | LAG$_8$ | RAE$_2$ | RAE$_4$ | RAE$_8$ | ICV |
|---|---|---|---|---|---|---|---|---|---|---|---|---|---|
| **TempSeq** | 40.32 | 25.81 | 24.19 | 19.89 | 18.28 | 17.74 | 30.11 | 25.27 | 24.73 | 26.61 | 23.39 | 25.23 | 24.54 |
| **DisambQA** | 71.51 | 67.20 | 61.83 | 63.44 | 68.82 | 68.28 | 59.68 | 72.58 | 74.73 | 64.68 | 63.30 | 65.14 | 65.45 |
| **DateUnd** | 58.06 | 54.84 | 52.15 | 54.30 | 55.38 | 58.60 | 53.23 | 53.23 | 56.99 | 55.96 | 56.42 | 56.42 | 58.00 |
| **TrackObj3** | 32.26 | 38.71 | 37.63 | 32.26 | 34.41 | 35.48 | 39.25 | 40.86 | 34.95 | 38.99 | 37.61 | 33.94 | 39.45 |
| **PengTable** | 34.15 | 37.80 | 36.59 | 39.02 | 35.37 | 34.15 | 34.15 | 36.59 | 35.37 | 36.84 | 41.23 | 41.23 | 38.23 |
| **GeomShapes** | 48.39 | 52.69 | 53.23 | 55.91 | 49.46 | 65.05 | 45.16 | 48.92 | 57.53 | 53.21 | 54.59 | 57.80 | 52.35 |
| **Snarks** | 66.67 | 60.53 | 57.89 | 54.39 | 50.88 | 46.49 | 57.02 | 62.28 | 51.75 | 58.90 | 59.59 | 54.79 | 60.53 |
| **RuinNames** | 67.39 | 55.43 | 52.72 | 54.35 | 54.35 | 47.83 | 56.52 | 62.50 | 55.98 | 54.17 | 55.09 | 50.93 | 66.34 |
| **TrackObj7** | 6.99 | 8.60 | 10.75 | 11.29 | 11.29 | 10.22 | 10.75 | 10.22 | 13.44 | 12.39 | 11.47 | 10.55 | 5.23 |
| **TrackObj5** | 13.44 | 15.05 | 17.20 | 18.28 | 15.05 | 14.52 | 14.52 | 13.98 | 15.59 | 16.51 | 16.06 | 16.06 | 14.98 |
| **LogDed3** | 51.08 | 51.08 | 44.62 | 50.00 | 50.00 | 51.61 | 49.46 | 54.30 | 57.53 | 51.38 | 56.42 | 53.67 | 52.45 |
| **Hyperbaton** | 59.68 | 74.73 | 70.43 | 72.04 | 67.74 | 64.52 | 59.14 | 63.44 | 68.82 | 61.47 | 57.80 | 69.72 | 63.44 |
| **LogDed5** | 33.33 | 38.71 | 38.17 | 37.63 | 38.71 | 38.71 | 38.17 | 39.25 | 40.32 | 36.24 | 40.37 | 37.61 | 35.53 |
| **LogDed7** | 29.03 | 32.80 | 26.34 | 26.88 | 29.57 | 25.81 | 32.26 | 27.42 | 27.96 | 27.52 | 31.19 | 32.57 | 29.57 |
| **MovieRec** | 76.76 | 87.03 | 85.95 | 83.78 | 85.95 | 88.65 | 80.54 | 87.03 | 85.41 | 82.03 | 86.64 | 83.41 | 86.34 |
| **SalTransErrDet** | 36.56 | 30.65 | 35.48 | 32.26 | 31.18 | 33.33 | 36.02 | 31.18 | 32.26 | 35.78 | 33.49 | 32.11 | 32.45 |
| **ReasColObj** | 35.48 | 33.87 | 28.49 | 27.96 | 33.87 | 28.49 | 32.26 | 32.26 | 29.03 | 34.40 | 31.65 | 27.98 | 31.65 |
| **Average** | 44.77 | 45.03 | 43.16 | 43.16 | 42.96 | 42.91 | 42.84 | 44.78 | 44.85 | 43.95 | 44.49 | 44.07 | 44.50 |

Table 10: Performance scores across tasks in BBH (Mistral-7B)

| Task Name | LARA | B-LARA | KATE$_2$ | KATE$_4$ | KATE$_8$ | ICL | LAG$_2$ | LAG$_4$ | LAG$_8$ | RAE$_2$ | RAE$_4$ | RAE$_8$ | ICV |
|---|---|---|---|---|---|---|---|---|---|---|---|---|---|
| abstract_algebra | 27.78 | 30.56 | 22.22 | 33.33 | 30.56 | 30.56 | 27.78 | 36.11 | 30.56 | 33.33 | 30.56 | 36.11 | 30.21 |
| anatomy | 59.15 | 60.56 | 61.97 | 60.56 | 56.34 | 57.75 | 57.75 | 59.15 | 57.75 | 57.75 | 61.97 | 60.56 | 55.20 |
| astronomy | 67.05 | 67.05 | 57.95 | 64.77 | 65.91 | 63.64 | 64.77 | 65.91 | 63.64 | 69.32 | 65.91 | 68.18 | 63.52 |
| business_ethics | 50.00 | 50.00 | 50.00 | 50.00 | 50.00 | 47.22 | 41.67 | 41.67 | 41.67 | 41.67 | 44.44 | 44.44 | 50.00 |
| clinical_knowledge | 69.00 | 70.50 | 68.50 | 71.50 | 73.50 | 70.00 | 70.50 | 72.00 | 70.50 | 69.00 | 70.00 | 71.00 | 70.50 |
| college_biology | 76.25 | 77.50 | 73.75 | 75.00 | 78.75 | 73.75 | 73.75 | 75.00 | 75.00 | 71.25 | 73.75 | 73.75 | 77.50 |
| college_chemistry | 58.33 | 61.11 | 55.56 | 58.33 | 61.11 | 50.00 | 52.78 | 55.56 | 58.33 | 52.78 | 52.78 | 55.56 | 50.00 |
| college_computer_science | 41.67 | 38.89 | 52.78 | 47.22 | 41.67 | 36.11 | 47.22 | 50.00 | 50.00 | 44.44 | 44.44 | 55.56 | 48.33 |
| college_mathematics | 22.22 | 25.00 | 38.89 | 33.33 | 36.11 | 36.11 | 22.22 | 22.22 | 27.78 | 27.78 | 25.00 | 22.22 | 27.78 |
| college_medicine | 65.14 | 62.39 | 62.39 | 63.30 | 63.30 | 65.14 | 63.30 | 64.22 | 63.30 | 64.22 | 64.22 | 64.22 | 63.30 |
| college_physics | 39.47 | 39.47 | 36.84 | 34.21 | 26.32 | 28.95 | 31.58 | 28.95 | 42.11 | 34.21 | 36.84 | 31.58 | 36.00 |
| computer_security | 69.44 | 75.00 | 66.67 | 72.22 | 69.44 | 72.22 | 72.22 | 69.44 | 72.22 | 72.22 | 75.00 | 75.00 | 70.20 |
| conceptual_physics | 57.89 | 55.56 | 55.56 | 55.56 | 56.73 | 57.89 | 54.97 | 56.73 | 56.14 | 54.97 | 57.89 | 56.73 | 58.48 |
| econometrics | 50.00 | 46.00 | 58.00 | 44.00 | 40.00 | 40.00 | 38.00 | 42.00 | 44.00 | 40.00 | 36.00 | 46.00 | 46.00 |
| electrical_engineering | 62.96 | 64.20 | 61.73 | 58.02 | 62.96 | 61.73 | 60.49 | 59.26 | 58.02 | 58.02 | 58.02 | 60.49 | 60.43 |
| elementary_mathematics | 42.00 | 41.50 | 38.00 | 41.50 | 40.50 | 40.00 | 40.50 | 41.00 | 39.50 | 41.00 | 40.00 | 39.50 | 43.50 |
| formal_logic | 32.26 | 32.26 | 32.26 | 35.48 | 35.48 | 37.10 | 27.42 | 27.42 | 27.42 | 27.42 | 29.03 | 33.87 | 37.10 |
| global_facts | 36.11 | 38.89 | 27.78 | 33.33 | 38.89 | 36.11 | 44.44 | 33.33 | 33.33 | 41.67 | 33.33 | 30.56 | 30.56 |
| high_school_biology | 75.00 | 74.50 | 77.00 | 75.00 | 76.50 | 74.50 | 74.00 | 75.00 | 76.00 | 74.00 | 67.00 | 72.00 | 74.00 |

Table 11: Performance scores across tasks in MMLU (Mistral-7B) Part 1

| Task Name | LARA | B-LARA | KATE$_2$ | KATE$_4$ | KATE$_8$ | ICL | LAG$_2$ | LAG$_4$ | LAG$_8$ | RAE$_2$ | RAE$_4$ | RAE$_8$ | ICV |
|---|---|---|---|---|---|---|---|---|---|---|---|---|---|
| high_school_chemistry | 56.83 | 56.12 | 54.68 | 52.52 | 55.40 | 51.80 | 54.68 | 55.40 | 53.96 | 53.96 | 41.00 | 46.00 | 48.27 |
| high_school_computer_science | 63.89 | 63.89 | 63.89 | 66.67 | 63.89 | 55.56 | 58.33 | 58.33 | 61.11 | 55.56 | 58.33 | 61.11 | 58.33 |
| high_school_european_history | 77.23 | 77.23 | 78.22 | 79.21 | 79.21 | 77.23 | 80.20 | 78.22 | 78.22 | 77.23 | 79.21 | 78.22 | 77.23 |
| high_school_geography | 85.82 | 86.57 | 83.58 | 82.84 | 86.57 | 83.58 | 82.09 | 83.58 | 81.34 | 81.34 | 82.84 | 83.58 | 87.31 |
| high_school_government | 85.27 | 86.05 | 86.05 | 86.82 | 84.50 | 83.72 | 83.72 | 83.72 | 82.95 | 84.50 | 83.72 | 83.72 | 86.05 |
| high_school_macroeconomics | 62.00 | 62.00 | 63.50 | 62.50 | 63.00 | 62.50 | 60.00 | 60.00 | 62.50 | 59.00 | 61.50 | 62.00 | 65.50 |
| high_school_mathematics | 32.00 | 29.50 | 33.50 | 33.00 | 35.50 | 31.50 | 41.00 | 41.00 | 37.50 | 43.50 | 39.00 | 38.50 | 39.50 |
| high_school_microeconomics | 68.39 | 67.82 | 68.39 | 64.94 | 64.37 | 62.07 | 66.09 | 67.82 | 66.67 | 64.37 | 66.67 | 65.52 | 67.41 |
| high_school_physics | 34.48 | 36.78 | 36.78 | 28.74 | 31.03 | 34.48 | 36.78 | 32.18 | 32.18 | 34.48 | 34.48 | 33.33 | 33.68 |
| high_school_psychology | 80.50 | 80.50 | 80.00 | 81.50 | 81.50 | 80.50 | 80.50 | 81.00 | 80.50 | 80.50 | 80.50 | 80.00 | 80.50 |
| high_school_statistics | 55.92 | 57.89 | 58.55 | 54.61 | 55.92 | 53.29 | 54.61 | 52.63 | 51.32 | 51.97 | 51.32 | 53.95 | 53.92 |
| high_school_us_history | 82.14 | 82.14 | 81.43 | 80.00 | 79.29 | 80.00 | 82.14 | 82.14 | 80.00 | 81.43 | 82.86 | 80.71 | 84.29 |
| high_school_world_history | 77.46 | 78.03 | 78.03 | 75.72 | 77.46 | 80.35 | 78.03 | 78.03 | 78.61 | 78.61 | 78.03 | 78.03 | 86.13 |
| human_aging | 67.92 | 67.92 | 68.55 | 67.30 | 66.67 | 66.67 | 67.30 | 68.55 | 69.81 | 67.30 | 68.55 | 67.30 | 67.44 |
| human_sexuality | 71.64 | 70.15 | 77.61 | 74.63 | 71.64 | 76.12 | 76.12 | 74.63 | 76.12 | 77.61 | 76.12 | 73.13 | 70.15 |
| international_law | 77.19 | 77.19 | 71.93 | 78.95 | 80.70 | 80.70 | 71.93 | 73.68 | 77.19 | 75.44 | 75.44 | 77.19 | 77.21 |
| jurisprudence | 79.55 | 79.55 | 77.27 | 75.00 | 79.55 | 72.73 | 77.27 | 75.00 | 77.27 | 75.00 | 77.27 | 77.27 | 72.73 |
| logical_fallacies | 70.71 | 69.70 | 80.81 | 77.78 | 77.78 | 77.78 | 78.79 | 79.80 | 80.81 | 80.81 | 83.84 | 77.78 | 78.79 |
| machine_learning | 52.08 | 47.92 | 58.33 | 43.75 | 47.92 | 45.83 | 41.67 | 45.83 | 47.92 | 41.67 | 41.67 | 45.83 | 47.92 |

Table 12: Performance scores across tasks in MMLU (Mistral-7B) Part 2

| Task Name | LARA | B-LARA | KATE$_2$ | KATE$_4$ | KATE$_8$ | ICL | LAG$_2$ | LAG$_4$ | LAG$_8$ | RAE$_2$ | RAE$_4$ | RAE$_8$ | ICV |
|---|---|---|---|---|---|---|---|---|---|---|---|---|---|
| management | 79.49 | 79.49 | 79.49 | 84.62 | 82.05 | 87.18 | 87.18 | 87.18 | 84.62 | 87.18 | 87.18 | 87.18 | 87.18 |
| marketing | 85.88 | 86.47 | 84.71 | 84.71 | 86.47 | 88.24 | 86.47 | 86.47 | 86.47 | 85.88 | 85.29 | 86.47 | 88.24 |
| medical_genetics | 72.22 | 72.22 | 69.44 | 72.22 | 72.22 | 66.67 | 66.67 | 69.44 | 72.22 | 66.67 | 72.22 | 69.44 | 86.11 |
| miscellaneous | 83.00 | 83.00 | 82.50 | 79.00 | 83.00 | 81.00 | 80.50 | 80.50 | 81.00 | 81.00 | 81.00 | 82.00 | 80.50 |
| moral_disputes | 75.50 | 74.50 | 72.50 | 72.00 | 73.50 | 73.00 | 74.00 | 74.00 | 74.50 | 72.50 | 73.00 | 73.50 | 74.00 |
| moral_scenarios | 43.00 | 45.50 | 40.00 | 39.50 | 40.00 | 42.00 | 45.00 | 47.00 | 43.50 | 39.50 | 39.50 | 45.50 | 42.50 |
| nutrition | 72.50 | 72.50 | 74.00 | 71.50 | 74.00 | 74.00 | 74.00 | 73.00 | 73.00 | 76.00 | 73.00 | 73.50 | 76.50 |
| philosophy | 73.50 | 73.00 | 71.50 | 73.50 | 74.50 | 69.00 | 70.00 | 70.50 | 73.00 | 70.50 | 72.00 | 74.00 | 74.00 |
| prehistory | 66.50 | 67.00 | 67.50 | 67.00 | 68.50 | 68.50 | 67.50 | 68.50 | 67.00 | 68.50 | 69.50 | 70.00 | 68.50 |
| professional_accounting | 49.50 | 49.50 | 50.50 | 50.50 | 50.00 | 49.00 | 49.00 | 51.50 | 48.50 | 50.00 | 51.50 | 50.00 | 53.00 |
| professional_law | 51.50 | 51.00 | 52.00 | 53.00 | 51.50 | 51.50 | 49.00 | 49.00 | 51.50 | 51.00 | 50.50 | 52.50 | 46.50 |
| professional_medicine | 68.00 | 70.00 | 69.50 | 66.50 | 67.50 | 68.00 | 65.50 | 67.50 | 68.00 | 66.00 | 65.00 | 66.00 | 69.00 |
| professional_psychology | 72.50 | 72.00 | 69.50 | 72.50 | 72.00 | 70.00 | 69.00 | 71.00 | 70.50 | 69.50 | 69.50 | 70.00 | 69.00 |
| public_relations | 80.43 | 80.43 | 73.91 | 69.57 | 76.09 | 73.91 | 76.09 | 76.09 | 73.91 | 76.09 | 76.09 | 73.91 | 76.09 |
| security_studies | 69.61 | 71.27 | 72.93 | 72.38 | 70.72 | 74.03 | 74.59 | 72.93 | 74.03 | 73.48 | 74.03 | 71.27 | 69.69 |
| sociology | 89.78 | 90.51 | 89.05 | 89.78 | 89.05 | 88.32 | 90.51 | 90.51 | 90.51 | 90.51 | 91.24 | 90.51 | 88.32 |
| us_foreign_policy | 91.67 | 91.67 | 91.67 | 86.11 | 86.11 | 86.11 | 88.89 | 88.89 | 88.89 | 88.89 | 91.67 | 88.89 | 88.89 |
| virology | 50.98 | 50.98 | 53.92 | 55.88 | 56.86 | 50.98 | 53.92 | 53.92 | 53.92 | 52.94 | 53.92 | 53.92 | 51.96 |
| world_religions | 85.98 | 85.98 | 84.11 | 85.05 | 84.11 | 85.05 | 84.11 | 85.05 | 84.11 | 85.05 | 85.98 | 86.92 | 84.11 |
| average | 63.93 | 64.12 | 63.99 | 63.48 | 63.93 | 62.84 | 62.96 | 63.32 | 63.55 | 62.99 | 62.99 | 63.61 | 64.03 |

Table 13: Performance scores across tasks in MMLU (Mistral-7B) Part 2

| Task Name | LARA | B-LARA | $KATE_2$ | $KATE_4$ | $KATE_8$ | ICL | $LAG_2$ | $LAG_4$ | $LAG_8$ | $RAE_2$ | $RAE_4$ | $RAE_8$ | ICV |
|---|---|---|---|---|---|---|---|---|---|---|---|---|---|
| **TempSeq** | 25.27 | 19.89 | 18.82 | 25.27 | 16.13 | 17.20 | 19.89 | 17.74 | 16.67 | 20.18 | 18.81 | 15.14 | 14.52 |
| **DisambQA** | 65.59 | 67.74 | 55.38 | 61.83 | 65.05 | 73.66 | 67.74 | 68.28 | 65.59 | 61.47 | 65.60 | 68.35 | 67.74 |
| **DateUnd** | 48.92 | 54.30 | 38.17 | 43.55 | 40.32 | 40.32 | 54.30 | 52.69 | 48.39 | 49.54 | 47.71 | 48.62 | 54.84 |
| **TrackObj3** | 32.26 | 32.80 | 39.78 | 38.71 | 29.03 | 37.10 | 38.71 | 34.95 | 34.41 | 32.57 | 30.28 | 37.61 | 34.95 |
| **PengTable** | 42.68 | 45.12 | 40.24 | 50.00 | 53.66 | 43.90 | 42.68 | 46.34 | 41.46 | 38.60 | 40.35 | 49.12 | 43.90 |
| **GeomShapes** | 54.30 | 47.31 | 51.08 | 56.99 | 48.92 | 63.98 | 46.77 | 48.92 | 46.77 | 44.04 | 53.67 | 50.00 | 64.52 |
| **Snarks** | 55.26 | 58.77 | 58.77 | 53.51 | 48.25 | 50.88 | 57.89 | 57.02 | 52.63 | 56.85 | 60.27 | 58.90 | 59.65 |
| **RuinNames** | 36.96 | 31.52 | 25.00 | 27.17 | 27.17 | 28.26 | 38.59 | 31.52 | 30.43 | 31.02 | 29.17 | 25.93 | 36.96 |
| **TrackObj7** | 12.90 | 11.83 | 15.05 | 14.52 | 13.98 | 12.90 | 12.37 | 15.59 | 10.75 | 5.96 | 11.93 | 13.76 | 11.29 |
| **TrackObj5** | 17.20 | 15.05 | 17.20 | 13.44 | 14.52 | 20.97 | 16.67 | 18.82 | 20.43 | 17.43 | 15.14 | 18.81 | 17.74 |
| **LogDed3** | 39.25 | 46.77 | 37.10 | 40.86 | 40.32 | 43.55 | 50.54 | 49.46 | 43.55 | 49.08 | 45.87 | 43.12 | 52.69 |
| **Hyperbaton** | 68.82 | 66.13 | 59.14 | 56.99 | 55.38 | 46.77 | 65.05 | 65.05 | 58.06 | 61.93 | 55.50 | 61.93 | 60.75 |
| **LogDed5** | 32.80 | 34.95 | 30.65 | 27.96 | 20.97 | 23.12 | 40.86 | 33.87 | 26.88 | 33.03 | 33.49 | 28.44 | 41.40 |
| **LogDed7** | 41.94 | 46.24 | 19.89 | 21.51 | 17.74 | 17.20 | 43.01 | 30.11 | 24.73 | 39.45 | 33.94 | 22.48 | 32.26 |
| **MovieRec** | 72.43 | 75.68 | 54.59 | 61.62 | 69.73 | 76.22 | 71.35 | 70.81 | 73.51 | 68.20 | 66.82 | 64.06 | 87.03 |
| **SalTransErrDet** | 28.49 | 30.65 | 29.03 | 29.03 | 30.11 | 0.00 | 32.26 | 34.41 | 32.26 | 30.73 | 38.99 | 28.90 | 0.00 |
| **ReasColObj** | 34.95 | 41.94 | 40.32 | 37.10 | 38.17 | 34.41 | 41.94 | 38.17 | 42.47 | 44.04 | 39.91 | 42.20 | 36.56 |
| **Average** | 41.77 | 42.75 | 37.07 | 38.83 | 37.03 | 37.08 | 43.57 | 41.99 | 39.35 | 40.24 | 40.44 | 39.85 | 42.16 |

Table 14: Performance scores across tasks in BBH (Gemma-7B)

| Task Name | LARA | B-LARA | KATE$_2$ | KATE$_4$ | KATE$_8$ | ICL | LAG$_2$ | LAG$_4$ | LAG$_8$ | RAE$_2$ | RAE$_4$ | RAE$_8$ | ICV |
|---|---|---|---|---|---|---|---|---|---|---|---|---|---|
| abstract_algebra | 27.78 | 25.00 | 22.22 | 33.33 | 25.00 | 27.78 | 25.00 | 33.33 | 41.67 | 30.56 | 30.56 | 30.56 | 25.00 |
| anatomy | 54.93 | 54.93 | 60.56 | 52.11 | 52.11 | 50.70 | 56.34 | 56.34 | 56.34 | 57.75 | 57.75 | 59.15 | 67.61 |
| astronomy | 73.86 | 71.59 | 71.59 | 68.18 | 64.77 | 69.32 | 72.73 | 75.00 | 75.00 | 75.00 | 72.73 | 73.86 | 69.32 |
| business_ethics | 52.78 | 55.56 | 61.11 | 55.56 | 58.33 | 58.33 | 52.78 | 55.56 | 55.56 | 52.78 | 50.00 | 52.78 | 66.67 |
| clinical_knowledge | 63.50 | 71.00 | 66.50 | 68.50 | 69.50 | 66.50 | 71.00 | 71.50 | 66.00 | 69.00 | 70.00 | 68.50 | 73.50 |
| college_biology | 75.00 | 75.00 | 80.00 | 73.75 | 75.00 | 70.00 | 76.25 | 76.25 | 73.75 | 72.50 | 77.50 | 75.00 | 81.25 |
| college_chemistry | 58.33 | 55.56 | 52.78 | 58.33 | 50.00 | 58.33 | 58.33 | 55.56 | 55.56 | 55.56 | 61.11 | 58.33 | 50.00 |
| college_computer_science | 38.89 | 50.00 | 50.00 | 47.22 | 38.89 | 36.11 | 50.00 | 52.78 | 52.78 | 52.78 | 44.44 | 33.33 | 55.56 |
| college_mathematics | 44.44 | 41.67 | 36.11 | 41.67 | 33.33 | 22.22 | 36.11 | 38.89 | 30.56 | 41.67 | 33.33 | 38.89 | 38.89 |
| college_medicine | 62.39 | 62.39 | 61.47 | 64.22 | 67.89 | 60.55 | 63.30 | 64.22 | 63.30 | 62.39 | 64.22 | 63.30 | 62.39 |
| college_physics | 34.21 | 31.58 | 34.21 | 42.11 | 34.21 | 42.11 | 34.21 | 28.95 | 28.95 | 31.58 | 31.58 | 36.84 | 42.11 |
| computer_security | 69.44 | 72.22 | 75.00 | 66.67 | 75.00 | 77.78 | 72.22 | 75.00 | 72.22 | 75.00 | 75.00 | 75.00 | 63.89 |
| conceptual_physics | 60.82 | 61.99 | 61.40 | 59.65 | 55.56 | 56.73 | 61.99 | 63.16 | 59.65 | 61.99 | 63.74 | 60.23 | 57.31 |
| econometrics | 48.00 | 48.00 | 52.00 | 46.00 | 42.00 | 50.00 | 46.00 | 46.00 | 44.00 | 42.00 | 42.00 | 46.00 | 56.00 |
| electrical_engineering | 59.26 | 65.43 | 61.73 | 56.79 | 56.79 | 51.85 | 55.56 | 59.26 | 64.20 | 59.26 | 58.02 | 61.73 | 70.37 |
| elementary_mathematics | 44.50 | 43.50 | 47.50 | 45.50 | 46.50 | 38.00 | 44.50 | 46.00 | 45.00 | 45.00 | 46.00 | 43.50 | 43.00 |
| formal_logic | 50.00 | 53.23 | 43.55 | 45.16 | 48.39 | 41.94 | 50.00 | 46.77 | 50.00 | 48.39 | 48.39 | 48.39 | 48.39 |
| global_facts | 36.11 | 41.67 | 38.89 | 44.44 | 33.33 | 30.56 | 41.67 | 44.44 | 36.11 | 36.11 | 38.89 | 30.56 | 25.00 |
| high_school_biology | 75.00 | 78.50 | 77.50 | 77.00 | 76.50 | 76.50 | 78.50 | 79.00 | 80.50 | 79.00 | 78.50 | 79.00 | 81.00 |

Table 15: Performance scores across tasks in BBH (Gemma-7B) Part 1

| Task Name | LARA | B-LARA | $KATE_2$ | $KATE_4$ | $KATE_8$ | ICL | $LAG_2$ | $LAG_4$ | $LAG_8$ | $RAE_2$ | $RAE_4$ | $RAE_8$ | ICV |
|---|---|---|---|---|---|---|---|---|---|---|---|---|---|
| **high_school_chemistry** | 56.83 | 51.80 | 53.96 | 56.83 | 48.20 | 48.92 | 56.12 | 53.24 | 53.96 | 53.96 | 53.96 | 57.55 | 57.55 |
| **high_school_computer_science** | 52.78 | 55.56 | 61.11 | 55.56 | 52.78 | 55.56 | 55.56 | 52.78 | 58.33 | 58.33 | 58.33 | 55.56 | 52.78 |
| **high_school_european_history** | 81.19 | 80.20 | 78.22 | 77.23 | 70.30 | 78.22 | 81.19 | 83.17 | 80.20 | 79.18 | 80.20 | 79.21 | 0.00 |
| **high_school_geography** | 84.33 | 87.31 | 84.33 | 85.82 | 85.82 | 82.84 | 85.82 | 83.58 | 82.09 | 86.57 | 86.57 | 83.58 | 88.81 |
| **high_school_government** | 86.82 | 86.82 | 89.92 | 86.82 | 86.82 | 85.27 | 86.05 | 86.05 | 87.60 | 87.60 | 86.82 | 88.37 | 89.15 |
| **high_school_macroeconomics** | 62.00 | 64.00 | 60.00 | 63.00 | 62.00 | 59.00 | 61.50 | 63.00 | 64.50 | 62.00 | 63.50 | 66.00 | 69.00 |
| **high_school_mathematics** | 42.50 | 44.00 | 35.50 | 32.50 | 33.50 | 26.00 | 42.50 | 39.00 | 39.50 | 36.00 | 36.00 | 38.00 | 40.00 |
| **high_school_microeconomics** | 67.82 | 70.11 | 68.39 | 68.39 | 70.11 | 65.52 | 70.11 | 68.97 | 66.67 | 68.39 | 68.39 | 65.52 | 71.26 |
| **high_school_physics** | 35.63 | 36.78 | 31.03 | 39.08 | 33.33 | 35.63 | 35.63 | 35.63 | 40.23 | 36.78 | 40.23 | 36.78 | 41.38 |
| **high_school_psychology** | 84.00 | 83.50 | 84.00 | 84.00 | 85.50 | 81.00 | 83.50 | 82.50 | 82.00 | 83.00 | 83.00 | 84.00 | 82.50 |
| **high_school_statistics** | 54.61 | 57.89 | 54.61 | 57.24 | 58.55 | 48.03 | 59.87 | 60.53 | 56.58 | 54.21 | 55.92 | 57.89 | 60.53 |
| **high_school_us_history** | 86.43 | 85.00 | 82.86 | 80.00 | 82.86 | 82.86 | 85.00 | 84.29 | 84.29 | 85.00 | 82.86 | 84.29 | 87.14 |
| **high_school_world_history** | 82.66 | 85.55 | 84.39 | 81.50 | 82.66 | 84.39 | 84.97 | 84.39 | 84.97 | 84.97 | 84.39 | 85.55 | 86.71 |
| **human_aging** | 72.33 | 72.33 | 67.92 | 72.33 | 74.84 | 69.81 | 72.33 | 73.58 | 74.21 | 72.96 | 72.96 | 72.33 | 68.55 |
| **human_sexuality** | 71.64 | 70.15 | 70.15 | 65.67 | 67.16 | 64.18 | 67.16 | 70.15 | 64.18 | 70.15 | 70.15 | 62.69 | 71.64 |
| **international_law** | 82.46 | 78.95 | 78.95 | 78.95 | 77.19 | 78.95 | 84.21 | 80.70 | 80.70 | 80.70 | 80.70 | 78.95 | 84.21 |
| **jurisprudence** | 70.45 | 75.00 | 72.73 | 68.18 | 70.45 | 63.64 | 75.00 | 70.45 | 68.18 | 75.00 | 68.18 | 79.55 | 70.45 |
| **logical_fallacies** | 73.74 | 73.74 | 76.77 | 77.78 | 77.78 | 71.72 | 73.74 | 74.75 | 73.74 | 73.74 | 72.73 | 77.78 | 76.77 |
| **machine_learning** | 54.17 | 50.00 | 52.08 | 43.75 | 47.92 | 50.00 | 47.92 | 60.42 | 58.33 | 52.17 | 50.00 | 50.00 | 47.92 |

Table 16: Performance scores across tasks in BBH (Gemma-7B) Part 2

| Task Name | LARA | B-LARA | KATE$_2$ | KATE$_4$ | KATE$_8$ | ICL | LAG$_2$ | LAG$_4$ | LAG$_8$ | RAE$_2$ | RAE$_4$ | RAE$_8$ | ICV |
|---|---|---|---|---|---|---|---|---|---|---|---|---|---|
| management | 89.74 | 92.31 | 92.31 | 79.49 | 71.79 | 84.62 | 92.31 | 89.74 | 84.62 | 89.31 | 89.74 | 89.74 | 87.18 |
| marketing | 86.47 | 87.65 | 87.65 | 88.24 | 86.47 | 84.71 | 88.24 | 89.41 | 90.00 | 87.06 | 90.00 | 87.06 | 88.24 |
| medical_genetics | 72.22 | 69.44 | 69.44 | 72.22 | 77.78 | 75.00 | 69.44 | 72.22 | 72.22 | 72.22 | 72.22 | 75.00 | 88.89 |
| miscellaneous | 82.50 | 83.50 | 83.50 | 84.50 | 81.00 | 82.00 | 84.00 | 83.00 | 82.00 | 84.50 | 84.00 | 84.50 | 82.00 |
| moral_disputes | 71.00 | 73.50 | 69.00 | 69.00 | 70.50 | 65.50 | 73.00 | 73.00 | 73.00 | 72.50 | 72.00 | 72.00 | 78.50 |
| moral_scenarios | 30.50 | 35.50 | 25.00 | 25.00 | 25.00 | 39.00 | 40.50 | 41.00 | 42.50 | 35.50 | 41.00 | 39.00 | 41.50 |
| nutrition | 74.00 | 75.00 | 39.50 | 37.00 | 33.50 | 70.00 | 75.00 | 77.50 | 77.00 | 78.50 | 76.00 | 73.00 | 78.00 |
| philosophy | 72.00 | 73.50 | 24.50 | 20.50 | 20.00 | 69.00 | 74.00 | 72.50 | 71.50 | 73.00 | 72.50 | 72.50 | 75.50 |
| prehistory | 71.50 | 73.00 | 35.00 | 33.00 | 26.50 | 70.50 | 74.50 | 73.50 | 73.00 | 73.00 | 73.50 | 74.50 | 69.50 |
| professional_accounting | 52.50 | 52.50 | 30.50 | 35.50 | 31.00 | 47.50 | 54.00 | 52.50 | 46.50 | 50.50 | 50.50 | 49.00 | 48.00 |
| professional_law | 53.00 | 53.00 | 22.50 | 22.00 | 22.00 | 24.00 | 54.50 | 54.50 | 48.50 | 51.00 | 51.00 | 46.00 | 46.00 |
| professional_medicine | 68.50 | 66.00 | 20.50 | 20.50 | 20.50 | 57.00 | 68.50 | 66.50 | 64.00 | 65.50 | 62.00 | 59.50 | 69.50 |
| professional_psychology | 66.00 | 69.00 | 37.00 | 40.50 | 35.00 | 67.50 | 69.00 | 70.00 | 69.00 | 71.00 | 70.00 | 69.00 | 72.50 |
| public_relations | 69.57 | 65.22 | 47.83 | 45.65 | 50.00 | 67.39 | 69.57 | 69.57 | 71.74 | 67.39 | 67.39 | 69.57 | 73.91 |
| security_studies | 76.80 | 77.35 | 23.76 | 19.34 | 19.34 | 75.69 | 77.90 | 75.69 | 74.03 | 77.45 | 77.35 | 75.69 | 72.93 |
| sociology | 85.40 | 87.59 | 38.69 | 34.31 | 37.96 | 82.48 | 86.13 | 86.86 | 85.40 | 86.13 | 83.94 | 84.67 | 88.32 |
| us_foreign_policy | 88.89 | 94.44 | 36.11 | 38.89 | 27.78 | 83.33 | 91.67 | 88.89 | 88.89 | 88.89 | 91.67 | 91.67 | 80.56 |
| virology | 40.20 | 53.92 | 37.25 | 44.12 | 42.16 | 53.92 | 53.92 | 54.90 | 53.92 | 54.90 | 55.88 | 51.96 | 52.94 |
| world_religions | 85.98 | 86.92 | 46.73 | 48.60 | 36.45 | 85.98 | 88.79 | 87.85 | 87.85 | 87.85 | 85.98 | 90.65 | 86.92 |
| average | 64.36 | 65.56 | 56.28 | 55.78 | 54.13 | 61.44 | 65.61 | 65.87 | 65.04 | 65.18 | 65.01 | 64.80 | 64.99 |

Table 17: Performance scores across tasks in MMLU (Mistral-7B) Part 3

| Task Name | LARA | B-LARA | KATE$_2$ | KATE$_4$ | KATE$_8$ | ICL | LAG$_2$ | LAG$_4$ | LAG$_8$ | RAE$_2$ | RAE$_4$ | RAE$_8$ | ICV |
|---|---|---|---|---|---|---|---|---|---|---|---|---|---|
| **TempSeq** | 21.10 | 30.76 | 20.64 | 14.22 | 15.14 | 20.64 | 17.52 | 16.24 | 8.12 | 19.72 | 19.72 | 19.72 | 65.59 |
| **DisambQA** | 55.05 | 72.02 | 65.60 | 62.84 | 61.93 | 62.39 | 41.45 | 35.04 | 39.32 | 60.09 | 58.26 | 59.17 | 51.08 |
| **DateUnd** | 56.42 | 57.80 | 51.83 | 61.47 | 56.42 | 56.42 | 50.85 | 57.26 | 52.56 | 50.92 | 55.50 | 55.50 | 52.69 |
| **TrackObj3** | 26.61 | 29.36 | 31.65 | 33.03 | 32.57 | 33.94 | 33.76 | 35.47 | 33.33 | 35.32 | 35.32 | 30.28 | 27.96 |
| **PengTable** | 49.12 | 50.00 | 51.75 | 51.75 | 56.14 | 50.00 | 45.38 | 40.00 | 50.00 | 39.47 | 37.72 | 46.49 | 37.80 |
| **GeomShapes** | 47.71 | 39.91 | 47.71 | 49.08 | 48.17 | 64.22 | 28.21 | 48.72 | 40.17 | 32.11 | 40.37 | 38.99 | 44.09 |
| **Snarks** | 67.81 | 64.38 | 63.01 | 63.01 | 62.33 | 61.64 | 55.56 | 62.35 | 64.81 | 59.59 | 63.01 | 59.59 | 60.53 |
| **RuinNames** | 54.63 | 53.70 | 46.30 | 41.67 | 47.69 | 54.63 | 48.28 | 46.98 | 51.29 | 46.76 | 47.69 | 48.15 | 28.80 |
| **TrackObj7** | 14.68 | 11.01 | 11.47 | 13.30 | 11.93 | 11.47 | 12.82 | 13.25 | 13.68 | 13.30 | 13.30 | 12.84 | 21.51 |
| **TrackObj5** | 16.51 | 16.06 | 11.47 | 14.68 | 12.84 | 15.14 | 17.09 | 18.38 | 15.81 | 17.43 | 16.51 | 14.68 | 23.12 |
| **LogDed3** | 59.63 | 61.93 | 55.50 | 52.75 | 50.00 | 47.25 | 60.26 | 57.26 | 58.55 | 56.88 | 57.80 | 55.96 | 41.40 |
| **Hyperbaton** | 75.23 | 72.02 | 72.48 | 68.81 | 72.94 | 62.84 | 77.78 | 75.64 | 75.21 | 78.90 | 74.77 | 72.48 | 73.12 |
| **LogDed5** | 43.58 | 44.95 | 39.45 | 38.53 | 38.99 | 43.58 | 42.74 | 42.31 | 42.74 | 42.20 | 39.45 | 40.37 | 39.25 |
| **LogDed7** | 45.41 | 45.41 | 40.37 | 37.61 | 37.16 | 37.16 | 45.30 | 47.01 | 46.58 | 43.12 | 44.50 | 43.12 | 44.62 |
| **MovieRec** | 71.43 | 75.58 | 63.13 | 69.12 | 71.43 | 79.26 | 66.52 | 66.52 | 58.80 | 71.89 | 72.81 | 66.82 | 86.49 |
| **SalTransErrDet** | 44.95 | 44.95 | 28.90 | 36.24 | 36.70 | 38.07 | 39.32 | 42.31 | 38.46 | 38.07 | 42.20 | 41.28 | 33.87 |
| **ReasColObj** | 56.88 | 40.83 | 39.91 | 40.37 | 43.58 | 37.16 | 53.85 | 51.71 | 41.88 | 52.29 | 50.00 | 43.58 | 48.92 |
| **Average** | 47.46 | 47.69 | 43.60 | 44.03 | 44.47 | 45.64 | 43.33 | 44.50 | 43.02 | 44.59 | 45.23 | 44.06 | 45.93 |

Table 18: Performance scores across tasks in BBH (Llama3.1-8B)

| Task Name | LARA | B-LARA | KATE$_2$ | KATE$_4$ | KATE$_8$ | ICL | LAG$_2$ | LAG$_4$ | LAG$_8$ | RAE$_2$ | RAE$_4$ | RAE$_8$ | ICV |
|---|---|---|---|---|---|---|---|---|---|---|---|---|---|
| abstract_algebra | 33.33 | 36.11 | 27.78 | 25.00 | 36.11 | 22.22 | 41.67 | 38.89 | 36.11 | 38.89 | 47.22 | 36.11 | 25.00 |
| anatomy | 63.38 | 66.20 | 64.79 | 61.97 | 64.79 | 66.20 | 63.38 | 63.38 | 66.20 | 63.38 | 63.38 | 63.38 | 73.24 |
| astronomy | 71.59 | 72.73 | 70.45 | 72.73 | 71.59 | 69.32 | 76.14 | 75.00 | 73.86 | 75.00 | 72.73 | 75.00 | 68.18 |
| business_ethics | 55.56 | 63.89 | 63.89 | 61.11 | 69.44 | 27.78 | 55.56 | 63.89 | 55.56 | 55.56 | 61.11 | 66.67 | 61.11 |
| clinical_knowledge | 75.50 | 77.00 | 77.50 | 78.50 | 76.00 | 75.50 | 78.50 | 78.00 | 78.50 | 77.50 | 79.00 | 77.50 | 75.50 |
| college_biology | 76.25 | 77.50 | 78.75 | 78.75 | 76.25 | 77.50 | 78.75 | 77.50 | 75.00 | 77.50 | 76.25 | 77.50 | 78.75 |
| college_chemistry | 50.00 | 47.22 | 52.78 | 44.44 | 44.44 | 50.00 | 50.00 | 47.22 | 47.22 | 50.00 | 50.00 | 47.22 | 50.00 |
| college_computer_science | 52.78 | 50.00 | 52.78 | 50.00 | 47.22 | 58.33 | 44.44 | 47.22 | 44.44 | 50.00 | 47.22 | 52.78 | 50.00 |
| college_mathematics | 41.67 | 50.00 | 47.22 | 41.67 | 50.00 | 27.78 | 44.44 | 50.00 | 47.22 | 38.89 | 47.22 | 47.22 | 30.56 |
| college_medicine | 56.88 | 62.39 | 63.30 | 59.63 | 63.30 | 63.30 | 61.47 | 63.30 | 61.47 | 63.30 | 63.30 | 64.22 | 64.22 |
| college_physics | 42.11 | 39.47 | 44.74 | 44.74 | 31.58 | 50.00 | 44.74 | 39.47 | 44.74 | 36.84 | 39.47 | 44.74 | 44.74 |
| computer_security | 75.00 | 75.00 | 75.00 | 75.00 | 72.22 | 72.22 | 80.56 | 77.78 | 75.00 | 80.56 | 77.78 | 80.56 | 69.44 |
| conceptual_physics | 60.23 | 60.23 | 59.06 | 58.48 | 56.14 | 58.48 | 59.06 | 61.99 | 57.89 | 59.65 | 60.82 | 61.40 | 58.48 |
| econometrics | 48.00 | 56.00 | 46.00 | 50.00 | 48.00 | 54.00 | 52.00 | 56.00 | 52.00 | 46.00 | 48.00 | 54.00 | 52.00 |
| electrical_engineering | 62.96 | 74.07 | 69.14 | 71.60 | 66.67 | 65.43 | 69.14 | 69.14 | 67.90 | 69.14 | 69.14 | 70.37 | 74.07 |
| elementary_mathematics | 44.00 | 46.00 | 43.00 | 44.00 | 43.00 | 43.50 | 47.00 | 46.00 | 45.50 | 46.50 | 47.00 | 46.00 | 43.50 |
| formal_logic | 51.61 | 54.84 | 50.00 | 40.32 | 51.61 | 37.10 | 46.77 | 43.55 | 54.84 | 50.00 | 40.32 | 46.77 | 43.55 |
| global_facts | 30.56 | 36.11 | 33.33 | 41.67 | 44.44 | 30.56 | 38.89 | 38.89 | 36.11 | 36.11 | 30.56 | 38.89 | 33.33 |
| high_school_biology | 78.50 | 80.50 | 80.00 | 80.00 | 79.00 | 78.00 | 81.00 | 80.50 | 80.50 | 81.50 | 81.50 | 81.00 | 81.50 |

Table 19: Performance scores across tasks in MMLU (Llama3.1-8B) Part 1

| Task Name | LARA | B-LARA | KATE$_2$ | KATE$_4$ | KATE$_8$ | ICL | LAG$_2$ | LAG$_4$ | LAG$_8$ | RAE$_2$ | RAE$_4$ | RAE$_8$ | ICV |
|---|---|---|---|---|---|---|---|---|---|---|---|---|---|
| high_school_chemistry | 57.55 | 61.87 | 60.43 | 57.55 | 61.87 | 58.27 | 56.83 | 59.71 | 56.83 | 56.12 | 0.00 | 0.00 | 57.55 |
| high_school_computer_science | 63.89 | 61.11 | 63.89 | 72.22 | 58.33 | 58.33 | 61.11 | 58.33 | 63.89 | 61.11 | 63.89 | 61.11 | 58.33 |
| high_school_european_history | 78.22 | 77.23 | 75.25 | 74.26 | 75.25 | 77.23 | 77.23 | 77.23 | 76.24 | 77.23 | 78.22 | 77.23 | 78.22 |
| high_school_geography | 85.82 | 88.81 | 83.58 | 88.06 | 86.57 | 87.31 | 84.33 | 85.82 | 85.07 | 81.34 | 85.82 | 86.57 | 85.07 |
| high_school_government | 88.37 | 85.27 | 90.70 | 89.15 | 86.05 | 86.05 | 86.82 | 86.82 | 88.37 | 86.05 | 86.05 | 86.05 | 86.82 |
| high_school_macroeconomics | 69.00 | 69.50 | 63.50 | 65.50 | 67.50 | 70.50 | 70.00 | 70.50 | 68.50 | 68.00 | 68.00 | 68.50 | 69.00 |
| high_school_mathematics | 36.50 | 43.50 | 42.00 | 39.00 | 42.50 | 39.50 | 44.50 | 45.00 | 41.00 | 44.50 | 42.00 | 41.50 | 44.00 |
| high_school_microeconomics | 71.26 | 75.29 | 70.11 | 71.26 | 72.99 | 72.41 | 71.26 | 72.99 | 74.71 | 72.41 | 74.14 | 75.86 | 72.99 |
| high_school_physics | 48.28 | 52.87 | 49.43 | 45.98 | 49.43 | 43.68 | 50.57 | 50.57 | 45.98 | 48.28 | 49.43 | 48.28 | 49.43 |
| high_school_psychology | 86.50 | 84.50 | 86.50 | 87.00 | 87.00 | 84.50 | 86.50 | 86.50 | 86.00 | 87.00 | 84.50 | 87.00 | 84.00 |
| high_school_statistics | 52.63 | 54.61 | 55.92 | 57.89 | 54.61 | 55.92 | 52.63 | 53.29 | 53.29 | 54.61 | 52.63 | 53.95 | 54.61 |
| high_school_us_history | 85.71 | 85.71 | 84.29 | 87.86 | 88.57 | 84.29 | 85.71 | 87.14 | 85.71 | 85.71 | 86.43 | 87.86 | 87.86 |
| high_school_world_history | 85.55 | 83.82 | 84.39 | 87.28 | 84.39 | 86.13 | 86.71 | 85.55 | 85.55 | 85.55 | 86.13 | 83.24 | 86.13 |
| human_aging | 71.70 | 71.70 | 68.55 | 69.18 | 71.70 | 70.44 | 69.18 | 69.81 | 69.81 | 67.30 | 70.44 | 72.33 | 71.07 |
| human_sexuality | 76.12 | 76.12 | 76.12 | 76.12 | 79.10 | 70.15 | 77.61 | 76.12 | 76.12 | 79.10 | 76.12 | 74.63 | 73.13 |
| international_law | 78.95 | 82.46 | 82.46 | 85.96 | 85.96 | 84.21 | 80.70 | 84.21 | 85.96 | 78.95 | 85.96 | 80.70 | 84.21 |
| jurisprudence | 63.64 | 65.91 | 72.73 | 68.18 | 68.18 | 72.73 | 70.45 | 65.91 | 68.18 | 70.45 | 68.18 | 68.18 | 75.00 |
| logical_fallacies | 77.78 | 78.79 | 79.80 | 78.79 | 78.79 | 78.79 | 78.79 | 79.80 | 75.76 | 78.79 | 77.78 | 79.80 | 80.81 |
| machine_learning | 43.75 | 39.58 | 37.50 | 45.83 | 43.75 | 47.92 | 37.50 | 37.50 | 43.75 | 43.75 | 43.75 | 45.83 | 47.92 |

Table 20: Performance scores across tasks in MMLU (Llama3.1-8B) Part 2

| Task Name | LARA | B-LARA | KATE$_2$ | KATE$_4$ | KATE$_8$ | ICL | LAG$_2$ | LAG$_4$ | LAG$_8$ | RAE$_2$ | RAE$_4$ | RAE$_8$ | ICV |
|---|---|---|---|---|---|---|---|---|---|---|---|---|---|
| **management** | 89.74 | 87.18 | 79.49 | 82.05 | 82.05 | 89.74 | 84.62 | 87.18 | 89.74 | 84.62 | 87.18 | 89.74 | 89.74 |
| **marketing** | 92.94 | 90.59 | 91.18 | 90.59 | 91.18 | 88.24 | 89.41 | 90.00 | 92.94 | 90.00 | 90.59 | 90.59 | 90.59 |
| **medical_genetics** | 88.89 | 88.89 | 88.89 | 88.89 | 86.11 | 86.11 | 88.89 | 88.89 | 88.89 | 88.89 | 88.89 | 88.89 | 88.89 |
| **miscellaneous** | 81.50 | 82.00 | 80.50 | 80.50 | 84.50 | 83.50 | 82.00 | 83.00 | 82.50 | 82.00 | 82.00 | 82.50 | 83.50 |
| **moral_disputes** | 74.50 | 76.00 | 75.00 | 77.00 | 77.50 | 74.00 | 74.50 | 75.00 | 75.50 | 74.00 | 75.50 | 76.00 | 76.50 |
| **moral_scenarios** | 46.50 | 46.00 | 34.00 | 38.00 | 40.50 | 40.50 | 37.50 | 41.50 | 49.00 | 40.00 | 46.00 | 45.50 | 41.50 |
| **nutrition** | 79.00 | 80.50 | 77.00 | 78.00 | 77.50 | 76.50 | 79.00 | 80.50 | 79.50 | 79.00 | 77.00 | 79.50 | 77.50 |
| **philosophy** | 72.50 | 71.00 | 74.00 | 72.50 | 73.00 | 74.00 | 72.50 | 72.00 | 72.50 | 74.00 | 75.00 | 75.00 | 73.00 |
| **prehistory** | 70.00 | 68.50 | 69.00 | 70.50 | 72.50 | 68.50 | 69.50 | 70.00 | 70.00 | 68.00 | 67.00 | 71.00 | 69.50 |
| **professional_accounting** | 50.00 | 48.50 | 48.00 | 48.50 | 51.00 | 53.00 | 50.00 | 50.50 | 47.50 | 50.50 | 50.50 | 48.50 | 47.50 |
| **professional_law** | 53.00 | 55.50 | 53.50 | 54.50 | 52.50 | 46.50 | 52.50 | 53.50 | 56.00 | 54.50 | 53.00 | 55.50 | 52.00 |
| **professional_medicine** | 65.00 | 65.00 | 71.00 | 67.00 | 67.50 | 71.00 | 68.50 | 67.00 | 66.50 | 66.50 | 67.00 | 65.00 | 68.50 |
| **professional_psychology** | 70.50 | 72.50 | 72.00 | 73.50 | 76.00 | 69.00 | 72.50 | 74.00 | 74.00 | 73.00 | 73.50 | 73.50 | 71.00 |
| **public_relations** | 73.91 | 76.09 | 80.43 | 78.26 | 78.26 | 76.09 | 73.91 | 76.09 | 76.09 | 73.91 | 76.09 | 78.26 | 78.26 |
| **security_studies** | 74.59 | 76.24 | 74.59 | 75.14 | 76.80 | 75.69 | 77.90 | 76.80 | 75.14 | 76.24 | 77.35 | 75.69 | 74.59 |
| **sociology** | 91.97 | 91.24 | 89.05 | 87.59 | 88.32 | 88.32 | 91.24 | 91.97 | 91.97 | 91.24 | 91.97 | 91.97 | 88.32 |
| **us_foreign_policy** | 88.89 | 88.89 | 77.78 | 77.78 | 83.33 | 88.89 | 88.89 | 86.11 | 88.89 | 86.11 | 88.89 | 88.89 | 86.11 |
| **virology** | 53.92 | 51.96 | 51.96 | 52.94 | 52.94 | 51.96 | 52.94 | 53.92 | 51.96 | 52.94 | 52.94 | 52.94 | 51.96 |
| **world_religions** | 84.11 | 84.11 | 83.18 | 85.05 | 84.11 | 84.11 | 84.11 | 84.11 | 84.11 | 84.11 | 83.18 | 85.05 | 85.05 |
| **average** | 66.54 | 67.80 | 66.62 | 66.75 | 67.19 | 65.64 | 66.71 | 66.88 | 66.69 | 66.22 | 66.91 | 67.24 | 67.10 |

Table 21: Performance scores across tasks in MMLU (llama3.1-8B) Part 3

