# OpenReview forum: "Divide, Reweight, and Conquer: A Logit Arithmetic Approach for In-Context Learning"
_ICLR.cc/2025/Conference — Submitted to ICLR 2025_

### Official Review · Reviewer_DoTr · 2024-10-31

**Soundness:** 2
**Presentation:** 3
**Contribution:** 2
**Rating:** 5
**Confidence:** 3

**Summary:**

The paper presents the Logit Arithmetic Reweighting Approach (LARA) for enhancing in-context learning (ICL) in large language models (LLMs). LARA improves ICL performance by dividing long prompts into shorter, parallelizable inputs and using logit-based ensembling to optimize memory usage and accuracy. This method also includes a Binary LARA (B-LARA) variant, which constrains the weight of each subgroup to binary values, reducing memory needs and focusing on the most informative inputs. The experiments on the BBH and MMLU benchmarks demonstrate that LARA and B-LARA outperform baseline methods in both efficiency and accuracy across different models.

**Strengths:**

1. Clear Presentation: The methodology is explained thoroughly, making LARA and B-LARA easy to understand.
2. Practical Relevance: LARA’s memory and computational efficiency make it well-suited for real-world applications, including black-box models.
3. Comprehensive Experiments: Extensive experiments across multiple benchmarks and ablation studies validate the effectiveness of the proposed approach.

**Weaknesses:**

1. While the empirical results are strong, the paper lacks a deeper theoretical analysis explaining why the logit reweighting improves performance, which could strengthen the foundational understanding of the approach.
2. The proposed method achieves good results on the MMLU and BBH datasets; however, it appears applicable only to multiple-choice questions or those with single-token answers, as it relies on the exact next-token logits for prediction, suggesting limited generalization capability.

**Questions:**

My main concerns have been listed above. I look forward to the authors' response and am willing to reopen and adjust the score upward.

---

> ### Author Response · Authors · 2024-11-20
> **Response to Reviewer DoTr (Part 1)**
>
> ## Q1: While the empirical results are strong, the paper lacks a deeper theoretical analysis explaining why the logit reweighting improves performance, which could strengthen the foundational understanding of the approach.
>
> Thank you for your suggestion. Our methods work because of these main reasons.
>
>
> 1. Aggregating logits from multiple subgroups can reduce the expected variance of the output logits. This is also the foundation of ensemble learning and majority voting. Previous papers [1][2][3] have made some theoretical analysis of the effect of logit ensembles.
>
> 2. B-LARA and LARA can solve the problem of long-context by splitting many in-context learning examples into several different groups. Excessively long contexts will introduce some problems like decaying attention score[4] or out-of-distribution of pre-training data[5][6], even exceeding the input window of LLM.
>
>
> [1]Allen-Zhu, Zeyuan et al. “Towards Understanding Ensemble, Knowledge Distillation and Self-Distillation in Deep Learning.” ArXiv abs/2012.09816 (2020)
>
> [2]Fathullah, Yassir et al. “Logit-Based Ensemble Distribution Distillation for Robust Autoregressive Sequence Uncertainties.” ArXiv abs/2305.10384 (2023)
>
> [3]Chen, Zixiang et al. “Towards Understanding Mixture of Experts in Deep Learning.” ArXiv abs/2208.02813 (2022)
>
> [4]Su, Jianlin et al. “RoFormer: Enhanced Transformer with Rotary Position Embedding.” ArXiv abs/2104.09864 (2021)
>
> [5]Peng, Bowen et al. “YaRN: Efficient Context Window Extension of Large Language Models.” ArXiv abs/2309.00071 (2023)
>
> [6]Jin, Hongye et al. “LLM Maybe LongLM: Self-Extend LLM Context Window Without Tuning.” ArXiv abs/2401.01325 (2024)
>
>
> ## Q2: The proposed method achieves good results on the MMLU and BBH datasets; however, it appears applicable only to multiple-choice questions or those with single-token answers, as it relies on the exact next-token logits for prediction, suggesting limited generalization capability.
>
> Thank you for your valuable suggestion. To demonstrate the generalization capability of our methods, we have included GSM8K, a natural language generation task on math reasoning, as an evaluation dataset. We have updated the results in Sec. 5.1.
>
> | Model       | Llama-3-8B-Instruct (GSM8K) | LongChat-7B-V1.5-32K (GSM8K) | Vicuna-7B-V1.5-16K (GSM8K) |
> |-------------|--------------------------------|-------------------------------|----------------------------|
> | ICL         | 66.64                         | 9.93                          | 16.30                     |
> | EarlyStop   | 71.21                         | 11.14                         | 17.44                     |
> | StructICL   | 69.43                         | 11.25                         | 17.12                     |
> | FocusICL    | _71.89_                       | **12.28**                     | _17.74_                   |
> | B-LARA      | **73.86**                     | _12.23_                       | **18.12**                 |
>
>
> As shown in the table, our method, B-LARA, outperforms all the baseline methods across 2 out of 3 models on GSM8K. This highlights the effectiveness and adaptability of our approach beyond single-token or multiple-choice tasks, addressing your concerns regarding generalization.
>
> **We hope this can address your concern**

---

> > ### Author Response · Authors · 2024-11-24
> > **Any New Comments are Welcomed**
> >
> > Dear Reviewer DoTr,
> >
> > We sincerely appreciate your thorough review and the valuable suggestions and comments you provided for our paper. We have carefully considered each point and have addressed them in detail in our rebuttal.
> >
> > As the Author-Review Discussion period is drawing to a close with only two days remaining, we would like to ensure that all your concerns have been adequately addressed. If there are any questions or unresolved issues, we are eager to provide further clarification or make necessary revisions.
> >
> > Best regards,
> >
> > The Authors

---

### Official Review · Reviewer_ZHUH · 2024-11-01

**Soundness:** 3
**Presentation:** 3
**Contribution:** 3
**Rating:** 5
**Confidence:** 4

**Summary:**

The paper proposed a new approach LARA to enhance in-context learning in LLM. The devised framework divides the demonstration examples into several subgroups, and ensemble the logit results from different groups based on Covariance Matrix Adaptive Evolution Strategy. They also proposes a binary version of weight selection, which is equivalent to subgroup pruning, and only selects more informative groups as the contributing demonstration. The evaluation experiments show the effectiveness of the proposed methods compared to original ICL and other variations.

**Strengths:**

1. LARA enhances ICL with a non-gradient optimization prompting strategy, and it shows performance gain in different settings.
2. The division strategy can handle the situation of longer context naturally.
3. The paper is clearly write and easy to follow.

**Weaknesses:**

1. In weight optimization algorithm for B-LARA, it requires about 20 iterations to obtain the final weights, which may results in inefficiency. Although finally only a subset of groups is involved, the intermediate steps still consume a lot. And there is no ablation study on the impact of the iteration number.
2. The method is limited to the case that the vocabulary space of the target answer is already known.

**Questions:**

In Figure 4, when number of examples used is 16, LARA without weight re-weighting seems performs better than the one with reweighting, could you provide more insights. And how could you verify the effectiveness of the re-weighting strategy if so.

---

> ### Author Response · Authors · 2024-11-20
> **Response to Reviewer ZHUH(Part 1)**
>
> Thank you for your valuable feedback. We have carefully addressed each of your questions below. We hope that our responses provide sufficient clarity and help you reassess your evaluation of our work.
>
>
> ## Q1: In weight optimization algorithm for B-LARA, it requires about 20 iterations to obtain the final weights, which may results in inefficiency. Although finally only a subset of groups is involved, the intermediate steps still consume a lot.
>
> Thank you for the insightful observation. Our method is applied during the training phase, where the weight optimization for B-LARA initially requires approximately 20 iterations to determine the final weights. Once this optimization phase is complete, the inference stage incurs no additional computational overhead compared to standard ICL. This setup provides an efficiency trade-off: while the initial optimization involves multiple iterations, the subsequent inference stage is computationally efficient, ensuring that LARA and B-LARA remain practical for real-world applications.
>
> ## Q2: And there is no ablation study on the impact of the iteration number.
> Thanks for pointing this out! We test the effect of iteration number on BBH using Mistral-7b model, and please find the results as follows.
>
> | Iter Number | 5     | 10    | 15    | 20    | 25    |
> | ----------- | ----- | ----- | ----- | ----- | ----- |
> | LARA        | 43.41 | 43.67 | 44.46 | 44.77 | **44.79** |
> | B-LARA      | 43.40 | 43.70 | 45.02 | **45.03** | 45.02 |
>
> Both LARA and B-LARA achieve strong performance with minimal iteration requirements, consistently surpassing the ICL baseline of 42.91. Notably, with just 5 iterations, both methods already outperform ICL, showcasing their effectiveness even under limited iteration settings. By 15 iterations, B-LARA reaches 45.02, achieving our reported results (20 iterations) early on without the need for extensive iterations.
>
> ## Q3: The method is limited to the case that the vocabulary space of the target answer is already known.
>
> Thank you for your concern. To address this, we add a new experiment on a natural language generation task, GSM8K, for mathematical reasoning. We update the results to Table 2 in our updated version of the paper, which we have also pasted below for your reference. We have updated the results in Sec. 5.1.
>
> | Model       | Llama-3-8B-Instruct | LongChat-7B-V1.5-32K | Vicuna-7B-V1.5-16K |
> |-------------|-------------------------|-----------------------|---------------------|
> | ICL         | 66.64                  | 9.93                 | 16.30              |
> | EarlyStop   | 71.21                  | 11.14                | 17.44              |
> | StructICL   | 69.43                  | 11.25                | 17.12              |
> | FocusICL    | *71.89*                | **12.28**              | *17.74*            |
> | B-LARA      | **73.86**              | *12.23*            | **18.12**          |
>
> As shown in the table, B-LARA achieves stronger performance compared to earlier approaches such as ICL, EarlyStop, and StructICL. Notably, it outperforms FocusICL in 2 out of 3 models, despite not relying on hidden states. This demonstrates that our method can effectively handle generation tasks like GSM8K, even without prior knowledge of the target answer’s vocabulary space.
>
> ## Q4: In Figure 4, when number of examples used is 16, LARA without weight re-weighting seems performs better than the one with reweighting, could you provide more insights. And how could you verify the effectiveness of the re-weighting strategy if so.
>
> Thank you for your thoughtful question. Indeed, at 16 examples, it may appear that LARA without re-weighting slightly outperforms the re-weighted version in Figure 4. This may result from the increasing complexity of the search space for weight optimization with more subgroups. It inspired us to develop B-LARA, which addresses this limitation by constraining the weights to binary values. As a result, B-LARA consistently surpasses the non-reweighted version across various scenarios, showcasing the robustness of the re-weighting strategy.
>
> **We hope this can address your concern:)**

---

> > ### Author Response · Authors · 2024-11-24
> > **Any New Comments are Welcomed**
> >
> > Dear Reviewer ZHUH,
> >
> > We sincerely appreciate your thorough review and the valuable suggestions and comments you provided for our paper. We have carefully considered each point and have addressed them in detail in our rebuttal.
> >
> > As the Author-Review Discussion period is drawing to a close with only two days remaining, we would like to ensure that all your concerns have been adequately addressed. If there are any questions or unresolved issues, we are eager to provide further clarification or make necessary revisions.
> >
> > Best regards,
> >
> > The Authors

---

### Official Review · Reviewer_aLU1 · 2024-11-02

**Soundness:** 3
**Presentation:** 3
**Contribution:** 3
**Rating:** 5
**Confidence:** 4

**Summary:**

This paper presents a method to address the issues of performance degradation and quadratic computational costs in the context of many-shot in-context learning (ICL). By grouping demonstrations for parallel input and employing Logit Arithmetic for reweighting and concatenation, the authors cleverly leverage the effective information from demonstrations. This approach alleviates the resource consumption associated with large-shot inputs and enhances the performance of ICL to some extent.

**Strengths:**

1. The issues of performance degradation and quadratic computational costs in many-shot in-context learning (ICL) are currently hot research topics in the AI community. The approach presented in this paper, which involves Logit Arithmetic concatenation after grouping inputs, is both novel and intriguing. The experiments effectively validate the method's improvements on these problems.

2. The method is supported by a solid theoretical foundation. The design of validation and cross-entropy loss can be viewed as a form of non-gradient fine-tuning for special tasks in the ICL context. B-LARA represents a further optimization of the LARA method's weights. The paper provides detailed formulas and algorithmic derivations to substantiate these claims.

**Weaknesses:**

1. The use of Logit Arithmetic reweighting and concatenation to leverage effective information from demonstrations is indeed a good approach. However, I am curious about how the authors address the issue of output prefix inconsistency during the decoding process after grouping different demonstrations as inputs. For example, if the first token of the output differs under different shot inputs, how can the subsequent tokens be rigorously and effectively subjected to Logit Arithmetic reweighting? This concern arises because most of the ground truth in the datasets used in this paper consists of a single token.

2. In the "Related Work" section, should the authors discuss more relevant methods for addressing many-shot ICL, such as FocusICL、STRUCTICL? Additionally, would it be beneficial to include these methods as baselines for comparison?

3. ICL is a broadly applicable and effective paradigm; however, the benchmarks used in the experiments lack diversity. For instance, I am curious about the performance of the proposed method on mathematical reasoning problems, such as those found in MATH or GSM8K.

**Questions:**

As mentioned in the above weaknesses.

---

> ### Author Response · Authors · 2024-11-20
> **Response to Reviewer aLU1 (Part 1)**
>
> We sincerely appreciate your detailed comments. Below, we have provided comprehensive responses to all your questions. We hope our explanations address your concerns and encourage you to re-evaluate our work based on these clarifications.
>
>
> ## Q1: The use of Logit Arithmetic reweighting and concatenation to leverage effective information from demonstrations is indeed a good approach. However, I am curious about how the authors address the issue of output prefix inconsistency during the decoding process after grouping different demonstrations as inputs. For example, if the first token of the output differs under different shot inputs, how can the subsequent tokens be rigorously and effectively subjected to Logit Arithmetic reweighting? This concern arises because most of the ground truth in the datasets used in this paper consists of a single token.
>
> Thank you for your thoughtful question! Similar to traditional autoregressive decoding, our method generates one token at a time by computing a weighted average of logits. Once the token for the current position is determined, it is appended to the context and used to generate the next token. As in standard generation processes, the logits for each new token are recalculated based on the updated contexts.
>
> While this approach ensures consistent and accurate decoding, it may introduce computational overhead as the context grows. To mitigate this issue, we employ a Key-Value Cache mechanism, which stores and reuses key-value pairs computed during earlier decoding steps. This optimization allows the model to bypass redundant computations by leveraging previously calculated attention information. Instead of recomputing attention over the entire sequence for each new token, the model processes only the newly added token, significantly reducing computational overhead.
>
> ## Q2: In the "Related Work" section, should the authors discuss more relevant methods for addressing many-shot ICL, such as FocusICL、STRUCTICL? Additionally, would it be beneficial to include these methods as baselines for comparison?
>
> Thank you for the valuable suggestion. In the revised version, we have included a discussion of relevant methods, such as FocusICL and StructICL, in the "Related Work" section. All of these works represent significant contributions to the field, offering innovative strategies for improving many-shot in-context learning.
>
> Below, we provide comparison between our method and two more baselines you suggested: StructICL and FocusICL. Because of no open-source code for FocusICL, we re-use the results in the FocusICL paper and implement our method to follow their experiment. We use the GSM8K and ARC dataset from FocusICL paper for experiments. We have updated the results in Table 8.  We have also compared StructICL with our methods on the datasets used in our paper, and added the results to Table 1 (BBH and MMLU) and Table 3 (GoEmotion and TacRED) in our revised version of the paper.
>
>
> | Model       | Llama-3-8B-Instruct (ARC) | Llama-3-8B-Instruct (GSM8K) | LongChat-7B-V1.5-32K (ARC) | LongChat-7B-V1.5-32K (GSM8K) | Vicuna-7B-V1.5-16K (ARC) | Vicuna-7B-V1.5-16K (GSM8K) |
> |-------------|------------------------------|--------------------------------|----------------------------|-----------------------------|--------------------------|--------------------------|
> | ICL         | 90.00                       | 66.64                         | 62.43                     | 9.93                       | 77.11                   | 16.30                   |
> | EarlyStop   | 90.47                       | 71.21                         | 62.43                     | 11.14                      | 78.14                   | 17.44                   |
> | StructICL   | 90.70                       | 69.43                         | 64.05                     | 11.25                      | 78.05                   | 17.12                   |
> | FocusICL    | **91.02**                   | _71.89_                       | **64.55**                 | **12.28**                  | _78.51_                 | _17.74_                 |
> | B-LARA      | _90.89_                     | **73.86**                     | _64.27_                   | _12.23_                    | **78.79**               | **18.12**               |
>
>
> As shown in the table, our method demonstrates significant improvement over previous approaches such as EarlyStop and StructICL across diverse models and datasets. When compared to more advanced methods like FocusICL, our methods outperform it in 3 out of 6 settings. Notably, different from FocusICL and StructICL, our methods does not rely on hidden states, highlighting the generalizability and effectiveness of our method.

---

> ### Author Response · Authors · 2024-11-20
> **Response to Reviewer aLU1 (Part 2)**
>
> ## Q3: ICL is a broadly applicable and effective paradigm; however, the benchmarks used in the experiments lack diversity. For instance, I am curious about the performance of the proposed method on mathematical reasoning problems, such as those found in MATH or GSM8K.
>
> Thank you for your insightful suggestion. As mentioned in our response to the previous question, we have included results on GSM8K across three different models. These results highlight the generalization ability of our approach in handling various types of tasks. For the complete results, please refer to our response to Question 2.
>
> **We hope this can address your concern:)**

---

> > ### Author Response · Authors · 2024-11-24
> > **Any New Comments are Welcomed**
> >
> > Dear Reviewer aLU1,
> >
> > We sincerely appreciate your thorough review and the valuable suggestions and comments you provided for our paper. We have carefully considered each point and have addressed them in detail in our rebuttal.
> >
> > As the Author-Review Discussion period is drawing to a close with only two days remaining, we would like to ensure that all your concerns have been adequately addressed. If there are any questions or unresolved issues, we are eager to provide further clarification or make necessary revisions.
> >
> > Best regards,
> >
> > The Authors

---

### Official Review · Reviewer_oFex · 2024-11-04

**Soundness:** 2
**Presentation:** 2
**Contribution:** 3
**Rating:** 5
**Confidence:** 4

**Summary:**

This paper proposes LARA, a new framework that enhances in-context learning (ICL) by ensembling logits from multiple demonstrations, achieving accuracy gains without requiring any parameter updates. The authors also introduce Binary LARA (B-LARA), which further refines efficiency by selectively excluding less informative demonstrations. Experiments on BBH and MMLU benchmarks validate that both LARA and B-LARA outperform standard ICL.

**Strengths:**

1. The proposed logits re-ensemble provides partial evidence supporting the effectiveness of grouped ICL, serving as an additional rationale for future research on extending this approach toward long ICL.
2. The proposed Binary LARA can further enhance performance while reducing costs.

**Weaknesses:**

1. The proposed method does not effectively address or mitigate the disadvantages of the related work introduced in the introduction. For instance, while OpenAI's API provides some logits information, LARA fundamentally remains a white-box approach. Additionally, Binary LARA also carries the risk of losing critical information. As a result, the motivation for the method is unclear, and there is a lack of in-depth analysis regarding why LARA is effective.
2. The selected baseline does not effectively demonstrate the validity of the proposed method. The comparison baseline does not specifically address the issue of large-scale demonstrations. Additionally, other relevant grouping-based methods, such as [1] and [2], were not included in the comparison. Additionally, the main text uses RAE, while the table uses IRE.
3. LARA requires iterative computation of optimal weights for the model and dataset before inference, which introduces latency. When there are few samples to infer, or even just a single sample—common in real-world interactions with LLMs—the proportion of additional costs relative to the inference becomes particularly high.

  [1] Structured Prompting: Scaling In-Context Learning to 1,000 Examples.

  [2] Focused Large Language Models are Stable Many-Shot Learners.

**Questions:**

1. For the experiments in Section 5.5, please provide additional details. For instance, does decoding each token require re-feeding the entire prefix to the API? If so, this would result in significant costs.
2. Where are the results for different candidate numbers of groups in Table 1 and 3?

---

> ### Author Response · Authors · 2024-11-20
> **Response to Reviewer oFex (Part 1)**
>
> Thanks for your comments. We provide a point-by-point response to all your questions here. We sincerely hope our responses address your concerns, and hope you can reconsider your assessment of our work if you find our response satisfactory.
>
>
> ## Q1: The proposed method does not effectively address or mitigate the disadvantages of the related work introduced in the introduction. For instance, while OpenAI's API provides some logits information, LARA fundamentally remains a white-box approach.
>
> Previous studies ([1][2][3]) considered black-box optimization compatible with logit information use. Therefore, we adopt this established practice in our method.
>
>
> Besides, LARA only requires output logits and can be adapted to API-based models that provide top-k tokens (as demonstrated in our experiments with GPT-4o-mini in Table 5). Generally, we consider such models to be black-box models. Thus, LARA should be treated as a black-box model.
>
>
> [1]Sun, Tianxiang, et al. "Black-box tuning for language-model-as-a-service." International Conference on Machine Learning. PMLR, 2022.
>
> [2]Huang, Chengsong, et al. "Lorahub: Efficient cross-task generalization via dynamic lora composition." arXiv preprint arXiv:2307.13269 (2023).
>
> [3]Chen, Sizhe, et al. "Adversarial attack on attackers: Post-process to mitigate black-box score-based query attacks." Advances in Neural Information Processing Systems 35 (2022): 14929-14943.
>
> ## Q2(1): Binary LARA also carries the risk of losing critical information. As a result, the motivation for the method is unclear.
>
> Thank you for your insightful feedback.
>
> We would like to highlight that numerous example-selection methods have been explored in prior research, many of which utilize only a subset of the given examples to enhance final performance ([4][5][6][7]). Furthermore, the empirical results show that B-LARA consistently outperforms LARA. These results suggest that removing some of the examples does not compromise the final performance of ICL results.
>
> [4]Liu, Jiachang, et al. "What Makes Good In-Context Examples for GPT-3?." arXiv preprint arXiv:2101.06804 (2021).
>
> [5]Nguyen, Tai, et al. "In-context example selection with influences." arXiv preprint arXiv:2302.11042 (2023).
>
> [6]Zhang, Yiming, et al. "Active example selection for in-context learning." arXiv preprint arXiv:2211.04486 (2022).
>
> [7]Su, Hongjin, et al. "Selective annotation makes language models better few-shot learners." arXiv preprint arXiv:2209.01975 (2022).
>
> ## Q2(2): There is a lack of in-depth analysis regarding why LARA is effective.
>
> There are three primary reasons why our proposed LARA and B-LARA work.
> 1. Ensemble methods can indeed enhance performance by leveraging mechanisms similar to self-consistency[8], where combining multiple perspectives or predictions often leads to more robust final outcomes with improved accuracy and stability. In LARA, we split the given examples into several groups and merge the output logits, which are similar in essence.
>
> 2. This approach allows for the selection of examples that are most relevant and effective for downstream tasks. In our experiment, for example, B-LARA removes about 55% relatively "useless" examples to improve the performance. By doing so, it can exclude instances that might be less informative or even misleading, ensuring that the LLM focus on high-quality examples that support overall task performance.
>
> 3. B-LARA and LARA can solve the problem of long-context by splitting many in-context learning examples into several different groups. Excessively long contexts will introduce some problems like decaying attention score[9] or out-of-distribution of pre-training data[10][11], even exceeding the input window of LLM.
>
> Thank you again for your thoughtful feedback, and we hope this explanation provides a clearer understanding of the method's motivation.
>
> [8]Wang, Xuezhi et al. “Self-Consistency Improves Chain of Thought Reasoning in Language Models.” arXiv preprint arXiv:2203.11171 (2022).
>
> [9]Su, Jianlin et al. “RoFormer: Enhanced Transformer with Rotary Position Embedding.” ArXiv abs/2104.09864 (2021)
>
> [10]Peng, Bowen et al. “YaRN: Efficient Context Window Extension of Large Language Models.” ArXiv abs/2309.00071 (2023)
>
> [11]Jin, Hongye et al. “LLM Maybe LongLM: Self-Extend LLM Context Window Without Tuning.” ArXiv abs/2401.01325 (2024)

---

> ### Author Response · Authors · 2024-11-20
> **Response to Reviewer oFex (Part 2)**
>
> ## Q3: The selected baseline does not effectively demonstrate the validity of the proposed method. The comparison baseline does not specifically address the issue of large-scale demonstrations. Additionally, other relevant grouping-based methods, such as [1] and [2], were not included in the comparison. Additionally, the main text uses RAE, while the table uses IRE.
> Thank you for your suggestion. Below, we provide comparison between our method and two more baselines you suggested: StructICL and FocusICL. Because of no open-source code for FocusICL, we re-use the results in the FocusICL paper and implement our method to follow their experiment. We use the GSM8K and ARC dataset from FocusICL paper for experiments. We have updated the results in Table 8. We have also compared StructICL with our methods on the datasets used in our paper, and added the results to Table 1 (BBH and MMLU) and Table 3 (GoEmotion and TacRED) in our revised version of the paper.
>
>
> | Model       | Llama-3-8B-Instruct (ARC) | Llama-3-8B-Instruct (GSM8K) | LongChat-7B-V1.5-32K (ARC) | LongChat-7B-V1.5-32K (GSM8K) | Vicuna-7B-V1.5-16K (ARC) | Vicuna-7B-V1.5-16K (GSM8K) |
> |-------------|------------------------------|--------------------------------|----------------------------|-----------------------------|--------------------------|--------------------------|
> | ICL         | 90.00                       | 66.64                         | 62.43                     | 9.93                       | 77.11                   | 16.30                   |
> | EarlyStop   | 90.47                       | 71.21                         | 62.43                     | 11.14                      | 78.14                   | 17.44                   |
> | StructICL   | 90.70                       | 69.43                         | 64.05                     | 11.25                      | 78.05                   | 17.12                   |
> | FocusICL    | **91.02**                   | _71.89_                       | **64.55**                 | **12.28**                  | _78.51_                 | _17.74_                 |
> | B-LARA      | _90.89_                     | **73.86**                     | _64.27_                   | _12.23_                    | **78.79**               | **18.12**               |
>
>
> As shown in the table, our method demonstrates significant improvement over previous approaches such as EarlyStop and StructICL across diverse models and datasets. When compared to more advanced methods like FocusICL, our methods outperform it in 3 out of 6 settings. Notably, different from FocusICL and StructICL, our methods do not rely on hidden states, highlighting the generalizability and effectiveness of our method.
>
> Regarding RAE and IRE, we are sorry for the confusion. The paper we use as baseline is named `Rationale-Augmented Ensemble (RAE)`  while its algorithm is called `Input-rationale ensemble (IRE)`.
> We sincerely apologize for the inconsistent denotation. We have revised it in our paper.
>
> ## Q4: LARA requires iterative computation of optimal weights for the model and dataset before inference, which introduces latency. When there are few samples to infer, or even just a single sample—common in real-world interactions with LLMs—the proportion of additional costs relative to the inference becomes particularly high.
>
> Thanks for pointing this out! We agree that for single sample use cases, the computational overhead of weight optimization could be significant. We have provided a lightweight version of our method, LARA w/o reweighting across subgroups (directly setting $w_i=1/k$) and shown its performance in Figure 4. While LARA w/o reweighting cannot outperform LARA and B-LARA, it can still outperform standard ICL. In real-world interactions such as chatting with LLMs, users can choose from a trade-off between a lightweight version for immediate outputs, or using weight optimization for scenarios requiring higher accuracy.
>
> ## Q5: For the experiments in Section 5.5, please provide additional details. For instance, does decoding each token require re-feeding the entire prefix to the API? If so, this would result in significant costs.
>
> We agree that when using API services, LARA requires re-feeding the entire prefix to the API at each decoding step. This is constrained by the current API implementations that do not support caching previous key-value states between calls -- which locally deployed models can support. However, this will only affect long-form generation tasks, for many applications where ICL is commonly used ([1]), such as classification, multiple-choice questions, and other tasks requiring short answers (like those in BBH and MMLU benchmarks), our methods only need a few rounds of API calls.
>
> [1] Li, Tianle et al. “Long-context LLMs Struggle with Long In-context Learning.” ArXiv abs/2404.02060 (2024)

---

> > ### Author Response · Authors · 2024-11-20
> > **Response to Reviewer oFex (Part 3)**
> >
> > ## Q6: Where are the results for different candidate numbers of groups in Table 1 and 3?
> >
> > Thank you for your question. For baseline methods such as StructICL and retrieval-based approaches, the number of candidates is a hyperparameter that must be manually selected by the user. In contrast, LARA offers the advantage of automatically determining the optimal number of candidates by minimizing the validation loss from both halves of the in-context examples. We have clarified this point in the revised version of the paper in Sec.4.1 (Hyperparameter Setting).
> >
> > In response to your query about the effect of the candidate number of groups on our method, we show the results on BBH using Mistral-7b model as below.
> > | Method      | ICL   | LARA  | LARA_2 | LARA_4 | LARA_8 |
> > |-------------|-------|-------|------------------|------------------|------------------|
> > | Accuracy(%) | 42.91 | **44.77** | 44.37           | 44.56           | 44.52           |
> >
> > | Method      | ICL   | B-LARA | B-LARA_2 | B-LARA_4 | B-LARA_8 |
> > |-------------|-------|--------|--------------------|--------------------|--------------------|
> > | Accuracy(%) | 42.91 | **45.03**  | 44.89             | 44.67             | 44.85             |
> >
> > The accuracy scores of different hyperparameter choices remain robust and can surpass standard ICL, demonstrating the advantage of our approach in automatically searching for this parameter through the loss function.
> > Hope our explanation can solve your concern.

---

> ### Author Response · Authors · 2024-11-24
> **Any New Comments are Welcomed**
>
> Dear Reviewer oFex,
>
> We sincerely appreciate your thorough review and the valuable suggestions and comments you provided for our paper. We have carefully considered each point and have addressed them in detail in our rebuttal.
>
> As the Author-Review Discussion period is drawing to a close with only two days remaining, we would like to ensure that all your concerns have been adequately addressed. If there are any questions or unresolved issues, we are eager to provide further clarification or make necessary revisions.
>
> Best regards,
>
> The Authors

---

> > ### Comment · Reviewer_oFex · 2024-11-25
> >
> > Thank the authors for the response and additional experiments.

---

> > > ### Author Response · Authors · 2024-11-25
> > >
> > > We sincerely appreciate your thoughtful review of our paper and your valuable feedback. We are delighted that you found  the additional experiments.
> > >
> > > We hope that our responses adequately address your concerns. If you have any additional suggestions or questions, please do not hesitate to let us know. Once again, we appreciate your time and effort in reviewing our work.
> > >
> > > We hope you can reconsider your assessment of our work if you find our response satisfactory.
> > >
> > > Best Regards,
> > >
> > > The Authors

---

### Author Response · Authors · 2024-11-20
**General response**

Dear Reviewers:

We genuinely appreciate your constructive reviews and feedbacks, which helped us to improve our work. We would like to start by expressing our appreciation for the positive recognition of the strengths of our study, including:

- Our proposed LARA and B-LARA approaches are well-supported (`oFex`,`aLU1`), novel (`aLU1`), and of practical usage (`DoTr`).
- LARA and B-LARA improve ICL performance(`oFex` `aLU1` `ZHUH` `DoTr`) and efficiency(`aLU1` `ZHUH` `DoTr`) through logit ensembling and subgroup reweighting by focusing on the most informative demonstrations(`oFex` `aLU1` `DoTr`)
- The paper is well-organized and clearly written(`ZHUH`,`DoTr`).

We've responded individually to each reviewer's questions. To address your concerns and enhance our submission, we've incorporated your suggestions to conduct further experiments and provide additional results and analysis. Below is a summary of the key updates:

- Experiments on GSM8K (Sec. 5.1)
- New baseline about StructICL and FocusICL. (Table 1, Table 2, Table 3, Table 8)
- Other clarification and explanations (Sec. 6 Sec. 5.1).

---

> ### Author Response · Authors · 2024-11-27
> **Any New Comments are Welcomed**
>
> Dear reviewers,
>
> We sincerely appreciate your thorough review and the valuable suggestions and comments you provided for our paper. We have carefully considered each point and have addressed them in detail in our rebuttal.
>
> As the Author-Review Discussion period is drawing to a close with only two days remaining, we would like to ensure that all your concerns have been adequately addressed. If there are any questions or unresolved issues, we are eager to provide further clarification or make necessary revisions.
>
> Best regards,
>
> The Authors

---

> > ### Author Response · Authors · 2024-12-02
> >
> > Dear Reviewers,
> >
> > We sincerely appreciate your detailed review and the valuable feedback on our paper. We have carefully considered each of your comments and addressed them thoroughly in our rebuttal.
> >
> > As we approach the final day of the Author-Review Discussion period, we want to ensure that all your concerns have been fully addressed. If there are any remaining questions or unresolved points, we would be happy to provide further clarification or make necessary adjustments.
> >
> > Best regards,
> > The Authors

---

### Meta-Review · Area_Chair_BMyR · 2024-12-21

**Metareview:**

This paper introduces the Logit Arithmetic Reweighting Approach (LARA), a novel framework that enhances ICL for LLMs by using logit-based ensembling of multiple demonstrations while reducing memory requirements through parallelization. Additionally, it presents Binary LARA (B-LARA), which simplifies the search space by constraining weights to binary values, and shows that both LARA and B-LARA outperform baseline methods in accuracy and memory efficiency in experiments on BBH and MMLU. Despite its contributions, the reviewers' feedbacks suggest that this paper still has to be improved on: the inadequacy of baseline comparisons, the latency introduced by weight optimization, potential output prefix inconsistencies during decoding, the lack of diverse benchmarks, inefficiencies in the weight optimization algorithm, and the limitation of requiring a known vocabulary space for target answers.

**Additional Comments On Reviewer Discussion:**

The discussions are somewhat thorough.

---

### Decision · Program_Chairs · 2025-01-22

Reject